# Magmatic and thermally produced reactive phosphorus 3.2 billion years ago and its implications for early life

Abu Saeed Baidya [1] ✉, Michelle M. Gehringer [2], Cristian Savaniu[3], Christoph Heubeck[4] & Eva E. Stüeken [1]

Reduced and polymerized phosphorus species may have been crucial for the origin and early evolution of life, as they are more reactive and soluble than phosphate. Thermal processes could have produced these phosphorus species; however, the underlying mechanism is poorly constrained, and geological evidence of polymerized species in the Precambrian is so far absent. Here, we investigated contact-metamorphic rocks from the ca. 3.22 Ga Moodies Group (South Africa), where mafic dikes intruded into shallow-marine sediments. We provide evidence of magmatic phosphite (up to 2.85 ppm) and metamorphic polyphosphate (up to 39.3 ppm). Additional laboratory experiments suggest that carbon can facilitate the thermal production of polyphosphates and reduced phosphorus species, including phosphide, from less reactive minerals such as apatite and vivianite. We conclude that magmatic and thermal-metamorphic rocks could have provided soluble and reactive phosphorus species crucial for the origin and early evolution of life.

Phosphorus (P) is a key element for modern biological systems and has likely played an important role in the origin of life on our planet[1,2]. Unlike other major elements (C, H, O, N, S) required for the origin of life, P does not have a stable gaseous phase, is less abundant, and is locked in solid rocks in its most abundant form, phosphate (P(V)), where P has an oxidation state of +5). Furthermore, P(V) is only weakly reactive towards organic compounds, which could have hindered abiotic phosphorylation of biomolecules. The low solubility and low reactivity of P(V) thus raise questions on the journey of P from rock to life - a conundrum that has become known as the 'P-problem'[3].

Reduced and polymerized P species have been proposed to solve the 'P-problem'[1,2]. Polymerized P species (also known as condensed P), including pyrophosphate (PP(V)), triphosphate (PPP(V)), tetra- and other higher-order phosphates (PPPP(V)), and cyclophosphates such as trimetaphosphate (PPPc) are more reactive than unpolymerized P(V) and, therefore, may have facilitated phosphorylation on the prebiotic Earth. The simplest polyphosphates is dimer PP(V), which could have been a phosphorylating agent[4], a source of ATP-based metabolism, and a potential energy source for early life[2]. Similarly, cyclophosphates may phosphorylate several organic compounds including glyceric acid, sugars, amino acids, and nucleosides[5]. Thermal processes in metamorphic and magmatic environments have been proposed to produce polyphosphates, such as (1) dry-heating of sodium or

ammonium phosphate salt (e.g., $NaH_2PO_4/NH_4H_2PO_4$) and rare minerals (whitlockite $(Ca_{18}Mg_2H_2(PO_4)_{14})$, newberyite $(MgHPO_4 \cdot 3H_2O)$, struvite $(MgNH_4PO_4 \cdot 6H_2O)$, brushite $(CaHPO_4 \cdot 2H_2O)$, and amorphous Fe-phosphate at 80–600 °C, especially in the presence of urea and other organics or Fe-Cr-Ni-bearing minerals[6–10]; and (2) partial dissolution of $P_4O_{10}$ produced in high-temperature (>1200 °C) volcanic processes[11]. Volcanic fumaroles in modern-day Japan and 2.5–14 Myr-old contact-metamorphic rocks from the Levant region have been shown to contain polyphosphates[11,12]. However, there is so far no geological record of polyphosphates older than the Miocene, so the validity and importance of thermal or magmatic formation of polyphosphates in early Earth history is not known.

Reduced P species such as phosphite (P(III), oxidation state of 3+), hypophosphite (P(I), oxidation state of +1), and phosphonate (molecules with P-C bonds and P with a 3+ oxidation state) are more soluble than P(V) in the presence of bivalent metals such as Ca and Fe[8]. Furthermore, they are more reactive than P(V). For example, P(III) is ca. 1000 times more soluble than P(V) in natural fluids including seawater and more efficient than P(V) in forming organophosphorus compounds[1,8]. P(I) is even more reactive than P(III), and P-C compounds already contain P-C bonds, suggesting that these reduced P species may act as more efficient phosphorylating agents than P(V). Thermal processes have been proposed as important routes to

[1]School of Earth and Environmental Sciences, University of St Andrews, Queen's Terrace, St Andrews, United Kingdom. [2]Department of Microbiology, University of Kaiserslautern-Landau (RPTU), Gottlieb-Daimler Str, Kaiserslautern, Germany. [3]School of Chemistry, University of St Andrews, North Haugh, St Andrews, United Kingdom. [4]Department of Geosciences, Friedrich-Schiller University Jena, Jena, Germany. ✉e-mail: asb27@st-andrews.ac.uk

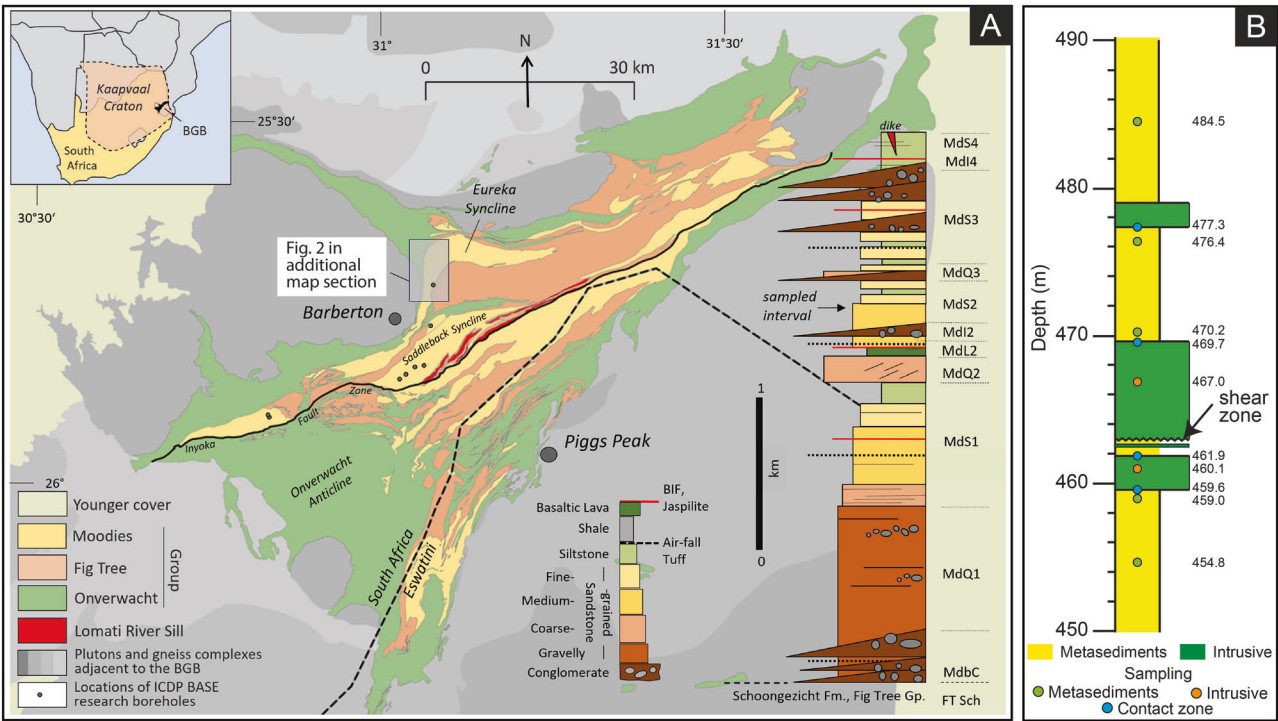

**Fig. 1 | Geological map of the study area and schematic diagram of the sampled core. A** Geological map of the Barberton Greenstone Belt (BGB), South Africa, modified from Homann et al.[60]. The Eureka Syncline is a large, refolded syncline in the north-central BGB. Stratigraphic column to the right shows generalized lithology of the Moodies Group in the Saddleback Syncline which preserves the greatest stratigraphic thickness of this unit. Gray box (dashed line) in the stratigraphic column shows the sampled section. See Heubeck et al.[21] for detailed locations of ICDP BASE boreholes. **B** Schematic stratigraphic column of the sampled section showing locations of the samples along with their specific depths. Note that the stratigraphy is overturned in core BASE 1A; drilling depth thus increases stratigraphically up. Supplementary Table S1 contains further information on sample positions.

form reduced P species, such as metamorphic heating of phosphate in the presence of $Fe^{2+}$, $H_2$ and/or organic matter[8,9,13–15]. However, these studies use sodium and ammonium phosphate as a precursor, which may not be relevant for early Earth as these salts do not readily form in nature[6]. Therefore, the underlying mechanism of P(V) reduction, particularly P(V) hosted in naturally occurring minerals such as apatite and vivianite that are considered less reactive, is not well understood. Importantly, only one study has documented P(V) reduction by metamorphic conditions in the Precambrian, specifically in seven Eoarchean carbonates and iron formation samples of amphibolite to granulite metamorphic grade[8,15]. Therefore, the importance of thermal P(V) reduction on the early Earth is so far poorly constrained.

To explore the effect of heat on P speciation in the Archean, we turned to the Paleoarchean Moodies Group (ca. 3.22 Ga) of the Barberton Greenstone Belt in South Africa, where mafic intrusions invaded into shallow-marine biomass-bearing sedimentary rocks[16,17], providing an ideal test bed for exploring if P polymerization and/or reduction could occur under Archean thermal metamorphic conditions. We collected five samples from metasedimentary units, two from intrusive units, and four from the contact zones between them and measured the concentrations of P(I), P(III), P(V), and PP(V), along with major and minor elemental abundances (see Methods). To validate our findings in Moodies Group and previous findings of phosphate reduction in other locations (e.g., the 2.5–14 Ma Levant region and 60 Ma Disko Island), we additionally performed laboratory experiments that replicated thermal metamorphism of biomass- and P-bearing sediments. We chose four prebiotically relevant P(V) minerals, namely, hydroxyapatite (($Ca_{10}(PO_4)_6(OH)_2$), magnesium phosphate ($Mg_3(PO_4)_2 \cdot xH_2O$);, vivianite ($Fe_3(PO_4)_2 \cdot 8H_2O$), and amorphous Fe-phosphate ($Fe_3(PO_4)_2 \cdot xH_2O$)[18] and two different C sources, namely carbon black (CB) and bacterial biomass (hereafter OM for 'organic matter'). We mixed the P(V) source, C source, and silica powder (mimicking very fine-

grained sand), pelleted them, and heated them up at 1150 °C or 1300 °C in an anoxic environment. The experimental products were analyzed by powder X-ray diffraction for phase identification. Phosphorus species were extracted from the experimental products using an Ethylenediaminete-traacetic acid-sodium hydroxide solution and measured using an IC-ICPMS[19] and nuclear magnetic resonance (NMR). Collectively, our data have implications for prebiotic P chemistry as well as for the origin and early evolution of life on our planet.

## Results and discussion
### Geological setting

We investigated the contact zones of Paleoarchean metasedimentary siliciclastic rocks intruded by mafic dikes in the north-central Barberton Greenstone Belt, South Africa (Fig. 1A). Samples were obtained from borehole BASE-1A, drilled within the framework of the ICDP BASE (Barberton Archean Surface Environments) project, which obtained continuous and unweathered core from the ca. 3.22 Ga Moodies Group (Fig. 1B, Supplementary Figs. S1-2). These are among the oldest well-preserved sedimentary rocks on Earth. The regional metamorphic grade is lower-greenschist facies with maximum temperatures of 420–460 °C, as indicated by Raman spectroscopy of carbonaceous matter[20].

Syn- and post-depositional magmatic activity affected Moodies Group strata. The most conspicuous contributor to this "flare-up" is the emplacement of the Moodies Lava, a ca. 20–400 m thick complex of basaltic amygdaloidal lavas approximately midway in the Moodies stratigraphic column, widely overlain by dacitic air-fall tuffs dated at 3219 ± 9 Ma, 3222 ± 8 Ma, and 3228 ± 8 Ma (LA-ICP-MS U-Pb ages of zircon[17]). An eruption age of about 3224 Ma is also consistent with a growing body of related age dates in the Moodies Group[21]. In stratigraphically correlatable Moodies Group strata in the central Barberton Greenstone Belt ca. 11 km to the south and southwest of the BASE-1A drill site, large mafic sills occur,

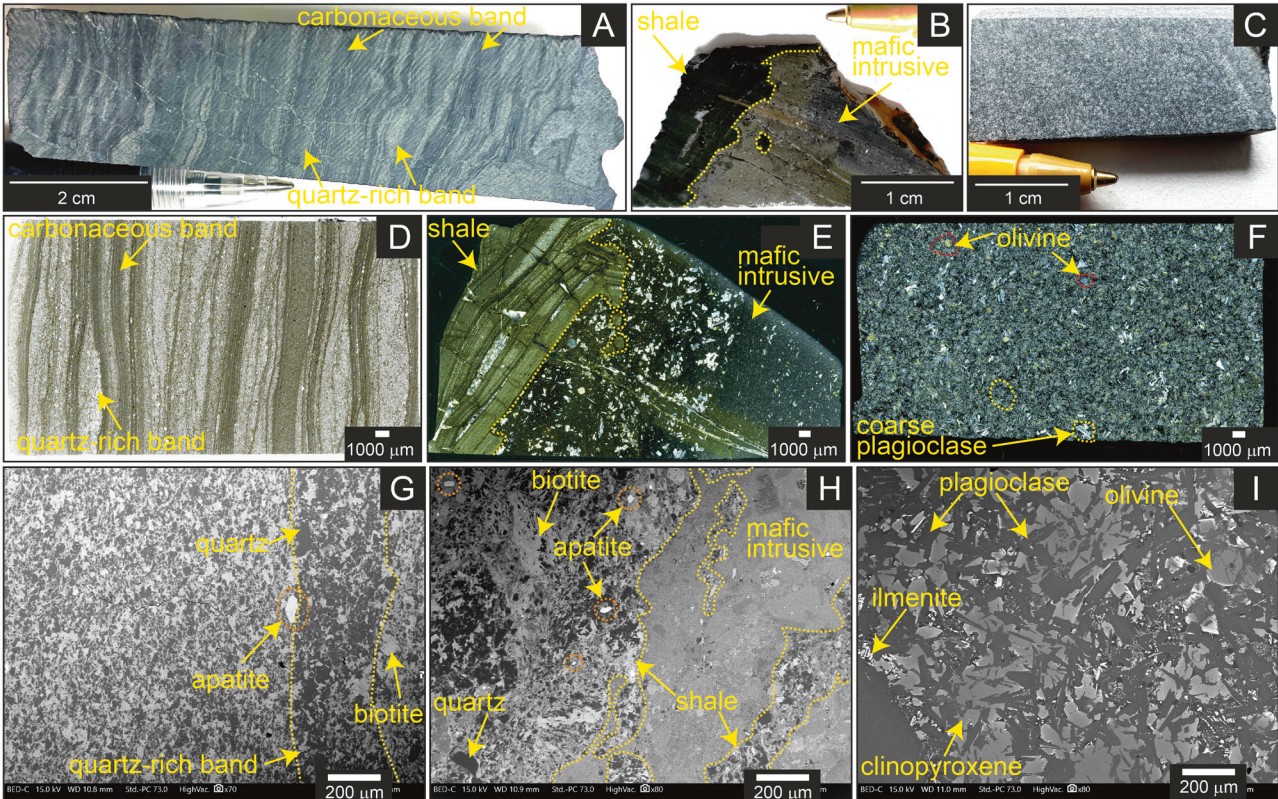

**Fig. 2 | Textural features of Moodies Group core BASE−1A. A–C, D–F, and G–I** are hand specimen, optical microscopic, and back-scattered electron images, respectively. **A, D,** and **G** (sample no. B20) show textures in metasedimentary strata. They contain apatite and lack visible alteration. **B, E,** and **H** (sample no. B13) illustrate textures of the contact zone and a close interaction between the intrusive and the metasedimentary strata. **C, F,** and **I** (sample no. B23) show textures of the intrusive unit, which shows no alteration. Sample depths are provided in Supplementary Table S1.

surrounded by halos of thermally altered Moodies sandstone and mafic stockwork dikes that apparently intruded in unconsolidated Moodies sands, resulting in local peperites[22]. Stratigraphic and geochronologic data indicate that stockwork dikes connect the sills, ca. 2 km below the paleosurface, with the Moodies lava, although no mappable connection has been documented yet. Stratiform mafic dikes, sills and thin lava flows were also encountered by borehole BASE-2A, ca. 7 km to the south of the BASE-1A location. Thermal alteration of sandstones was noted in borehole BASE-4B, ca. 14 km to the SSW. This alteration is probably the result of a nearby bedding-parallel-trending feldspar-porphyritic dike[21]. The diverse nature of (sub-)volcanic contributions to the Moodies Group sedimentary environments was summarized by ref. 16. The authors also documented from outcrop detail how magma at the base of the Moodies lava intruded downwards into fractured, apparently cemented Moodies sandstones (Fig. 13 in ref. 16). This location lies only ca. 500 m NE of and stratigraphically ca. 30 m below the projected surface location of the investigated samples. Furthermore, there are two younger thermal and magmatic events in the BGB: a thermal alteration related to sulfidic and Au mineralization at ca. 3084 Ma in the northern part of BGB[23] and a younger magmatic event at ca. 2967 Ma, comprising NW-SE oriented, intermediate, feldspar-porphyritic, dolerite dikes[24] (additional information is given in the Supplementary Material).

The borehole samples studied here are approximately 40 drilled meters (or ca. 25 stratigraphic meters) above the Moodies lava complex (Fig. 1). Core description shows that the contacts are intrusive; they are, in part, curved and free of indications of faulting or shearing (Supplementary Figs. S1 and S2). The thermal alteration event at ca. 3084 Ma is not associated with magmatism in the study area and the composition and structural orientation of the dolerite dikes of ca. 2967 Ma are different from the sampled intrusive rocks. All available geological evidence thus indicates that the contacts investigated here are likely part of magmatic-sedimentary interaction that took place during the Paleoarchean, prior to 3.2 Ga. Intrusions at ca. 3.224 Ga may have occurred at shallow depths and into unconsolidated sediment, as suggested by the nonlinear contacts, or occurred thousands to a few million years later, during late deformation of the BGB and subsequent beginning consolidation of the Kaapvaal craton.

## Textures and phosphorus speciation in Moodies Group rocks
The metasedimentary rocks from the Moodies Group investigated in this study show alternate banding with dark and light bands containing mostly biotite and quartz, respectively (Fig. 2A, D, G), indicating alternation between shale, siltstone, and fine-grained sandstone. K-feldspar, calcite, chlorite, and plagioclase are common while zircon, arsenopyrite, and apatite are accessory phases (Fig. 2G). The metasedimentary samples do not show alteration (Fig. 2D), although minor sericitization is observed at some places. The mafic intrusive bodies are massive and contain pyroxene, plagioclase, ilmenite, and olivine. Ilmenite shows skeletal textures and is mostly associated with fine-grained feldspar, pyroxene, and glassy material (Supplementary Fig. S3A, B), which is indicative of quenching[25]. These quenched spots contain rare microcrystalline apatite (Supplementary Fig. S3D). Olivine shows minor alteration along grain boundaries and fractures; however, the overall alteration in the rock is limited (Fig. 2F).

Of the four contacts zones that we studied, two are dominated by shale at the metasedimentary-igneous interface while two are dominated by siltstone (Supplementary Fig. S4). The intrusive units at the contact zone show variable degrees of sericitization, mostly along fractures (Fig. 2E), indicating some degree of fluid alteration. These samples show textural evidence of intrusion of a mafic melt into pre-existing sedimentary rock. First, fragments of laminated metasedimentary rock are incorporated into the intrusive units, indicating that the sedimentary rocks were already emplaced and at least somewhat lithified by the time the magma intruded

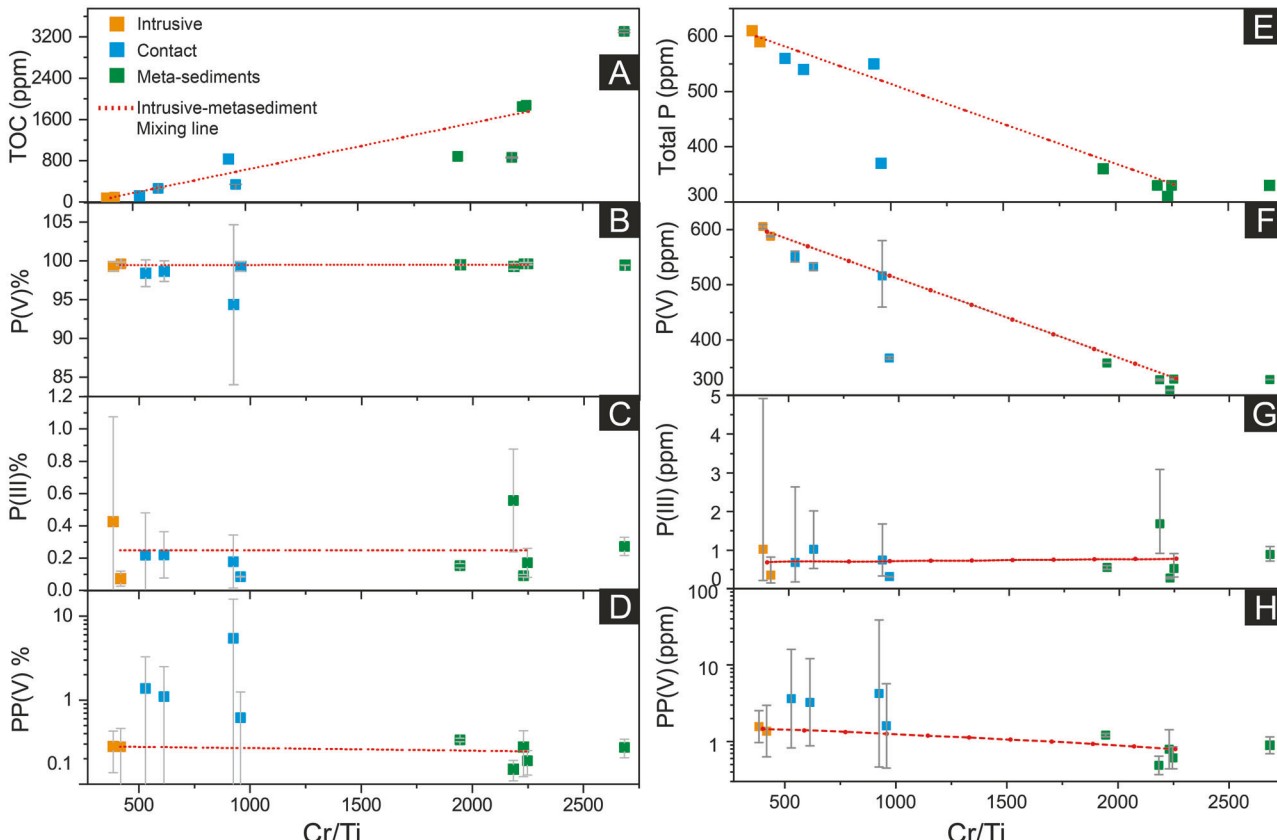

**Fig. 3 | Concentrations of different P species and total organic carbon (TOC) in studied samples from the Moodies BASE−1A borehole.** Concentrations of P(III), P(V), and PP(V) are measured with the IC-ICP-MS set-up while total P and TOC contents are measured by ICP-MS and IRMS, respectively. The P speciation data is presented in Table S3 in the Supplementary Material. The presented concentrations of different P species represent estimated concentrations in bulk samples. **A** Total organic carbon (TOC) concentrations are higher, as expected, in metasedimentary units than in intrusive and contact-zone samples. In **B**–**D** geometric means of P(III), P(V), and PP(V) are plotted due to large variations in the acquired data. Percentages of different P species in total extracted P (in EDTA-NaOH) are shown in **F**–**H**. These ratios are presented as averages and as their standard deviations (grey line). The

geometric mean is used here because that is appropriate for datasets which include ratios and where the data show relatively wide scatter. The arithmetic mean would be skewed towards higher values. This approach is thus more conservative. The red dotted lines represent conservative mixing lines of the intrusives and metasediments, calculated from ratios of Cr and Ti, which were fluid-immobile elements in the Archean[61] and whose ratios shows strong contrasts between the lithologies. Generally, where the contact zone sample data fall on that mixing line (e.g., P(III) in panel G), the data can be explained by simple physical mixing between magma and sediment; where points deviate from the line, additional chemical reactions occurred at the contact. **E** shows total P contents in the intrusive, metasedimentary, and contact zone samples.

(Fig. 2B, E). Second, we observe formation of cryptocrystalline pyrite and pyrite aggregates and compositional change along the intrusive side of the contact zone (Supplementary Fig. S5A). On the metasedimentary side, we observe aggregation of biotite (Supplementary Fig. S5B). Third, apatite is present at the contact between the metasedimentary unit and the intrusive bodies (Supplementary Fig. S5D). Collectively, these features indicate a thermal effect on a P(V) source and associated contact-metamorphic reactions.

Total Organic C concentrations are highest (833–3310 ppm) in metasedimentary rocks and lowest (76–93 ppm) in the intrusive bodies (Fig. 3A, Supplementary Table S2). Total P concentrations are higher (590–610 ppm) in the intrusive bodies compared to the metasedimentary rocks (310–330 ppm), although apatite is absent or rare in the former and common in the latter (Fig. 3A). Contact-zone samples have intermediate concentration of P (370–560 ppm), consistent with conservative mixing of the two units in bulk powders of the contact zones (Fig. 3E). Phosphorus speciation data for Moodies Group samples are given in Supplementary Table S3. We note that P extraction yields with EDTA-NaOH solutions are low, 1.4–2.4% for the metasedimentary unit, 5.2–5.6% for the intrusive unit, and 3.8–5.2% for the contact zone samples; however, these yields are consistent with previous studies[8,15]. The reported results are normalized to the extraction yields. Stronger solvents may increase the yield but risk losing P speciation via oxidation of reduced P or disintegration of polymerized P.

The low and variable yield thus likely explains the high level of uncertainty in measured P(III) and PP(V) concentrations, but general trends between the two lithologies and the contact zones are nevertheless comparable.

We estimated concentrations of three P species in the bulk Moodies samples using the ratios of them in the EDTA-NaOH extract, total P contents of the rocks, and extraction yields. The data of extracted and estimated concentrations are shown in Table S4. Extracted P(V) concentrations in the intrusive bodies, the metasedimentary unit, and the contact zones follow a similar pattern as total P concentrations (which were measured by bulk digestion of the rock, see Methods and Materials section), indicative of conservative mixing (Fig. 3F). Averages of estimated phosphite concentrations in intrusive, contact zone, and metasediments are up to 2.84, 1.64, and 1.13 ppm, respectively, which follows conservative mixing predictions with similar percentages of total extracted P (0.25, 0.25, and 0.18%, in metasedimentary, intrusive, and the contact zone samples, respectively; Fig. 3C, G). For PP(V), the highest estimated (2.32–39.22 ppm) concentrations are observed in the contact zone, followed by the intrusive bodies (1.65–1.65 ppm) and sedimentary rocks (0.47–1.20 ppm) (Table S5, Fig. 3H). The highest relative proportion of PP(V) in the EDTA-NaOH extract is observed in the contact zone samples (averages for metasedimentary, intrusive, and the contact zone samples are 0.25, 0.28, and 2.14%, respectively), which cannot be explained by conservative mixing between intrusive rocks and metasediments (Fig. 3D). The deviation from the mixing

**Fig. 4 | NMR spectra of end products from vivianite and amorphous Fe-phosphate experiments.** All the experiments were performed at 1150 °C except for the top Viv + SiO₂ + CB experiment which was done at 1300 °C. In general, the addition of carbon in either form, i.e., carbon black (CB) and organic biomass (OM), produced several reduced and polymerized phosphorus species. Minor shifts in P(V) and other peaks are due to minor differences in pH of the solution.

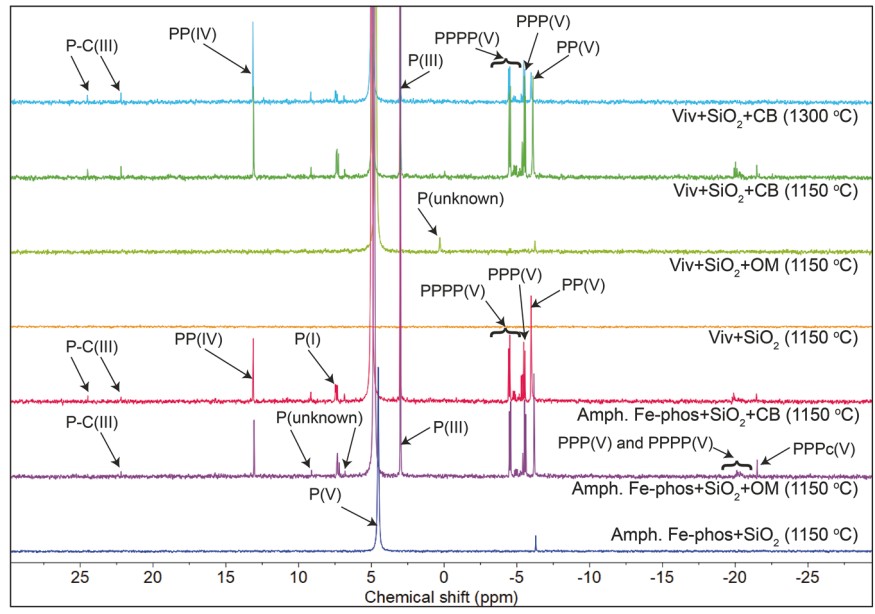

**Fig. 5 | NMR spectra of experimental products containing apatite.** Experiments were conducted at 1300 °C except for the bottom three experiments that were conducted at 1150 °C. Addition of an iron source to a mixture of apatite (Apa), silica (SiO₂), and carbon black (CB) produced several reduced and some polymerized P species.

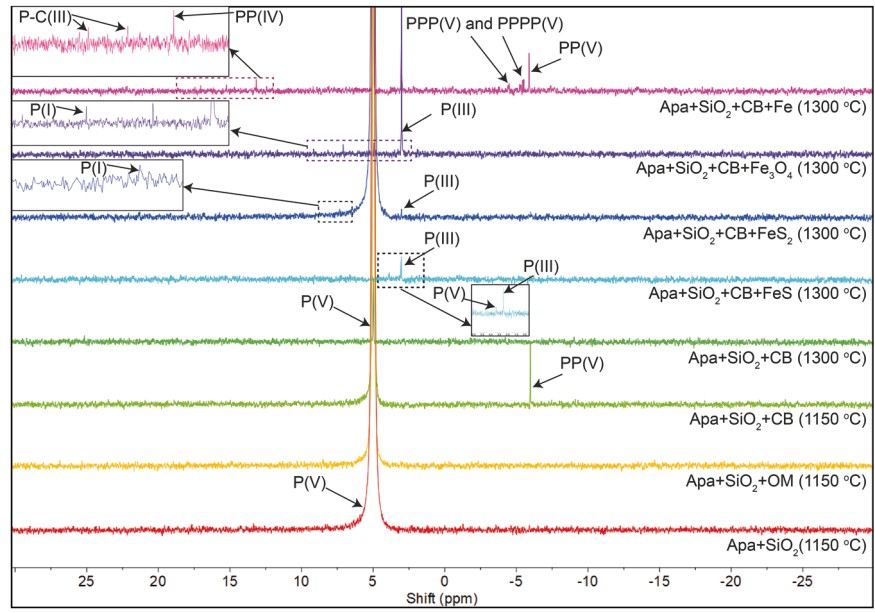

line suggests formation of additional PP(V) at the contact zones. We did not notice any trends in the metasedimentary rocks with increasing distance from the intrusions.

## Carbon-phosphate mineral heating experiments

To further explore the effects of heating on P-bearing sediments that also contain ferrous iron and carbon, we performed a series of experiments, which are listed in Supplementary Table S4 and results are summarized in Supplementary Table S5.

Three types of control experiments were conducted to verify the sources of polymerized and reduced P in starting materials. (1) Dry heating the silica powder alone did not produce polymerized or reduced P species but produced significant P(V) at 1150 °C suggesting that the silica powder may contain some P(V) either in the crystalline structure of silica or as a minor separate phase but most likely was sufficiently clean in terms of polymerized and reduced P species. (2) Heating a mixture of silica and OM without a P-source produced significant PP(V) (1.81% of total extracted P).

Similarly, a mixture of silica and CB produced significant amounts of P(I), P(III), PP(V), and PPP(V) (0.04, 0.48, 22.05, and 2.97% respectively). NMR analysis identified at least two organophosphate and one phosphonate compound in the OM but none in the CB. Therefore, a part of PP(V) in the heated OM and silica mixture may be produced due to polymerization of organophosphate compounds present in the OM, but collective data from these two control experiments point to the carbon-enhanced polymerization of the unidentified P(V)-phase present in the silica. The unheated mixture of silica and CB contained a similar amount of P(III) compared to the heated mixture, implying that the carbon sources contributed background levels of reduced P, but heating did not enhance P(III) production in these controls. (3) Heating phosphate precursors in the presence of silica, particularly magnesium phosphate, hydroxyapatite, and amorphous Fe-phosphate at 1150 °C, produced PP(V) with a yield of 0.003, 0.026, and 2.065%, respectively (Figs. 4, 5). Neither P(III), nor any other reduced P species were detected in any of these three control experiments. The vivianite and silica mixture produced P(III) and another unknown P species,

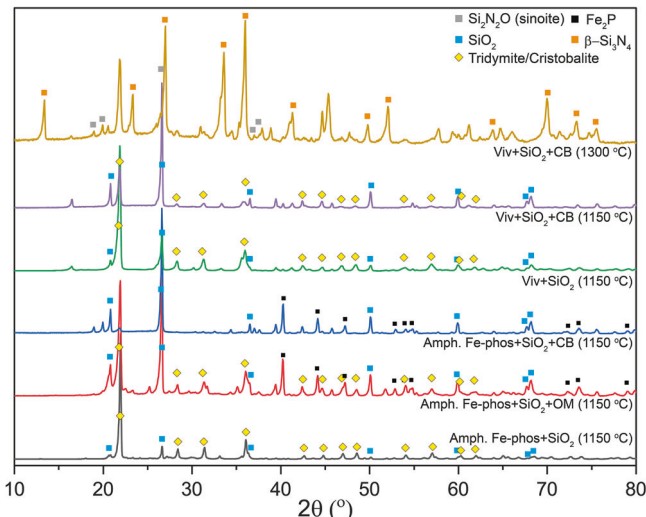

**Fig. 6 | XRD patterns of the products from amorphous Fe-phosphate and vivianite experiments.** Barringerite ($Fe_2P$) is detected in two amorphous Fe-phosphate-bearing experiments in the presence of carbon (organic biomass (OM) and carbon black (CB)). This phosphide is possibly present as a minor phase in the vivianite-CB-bearing experiments at 1150 and 1300 °C. Some phases remain unidentified (based on a comparison with entries in the Inorganic Crystal Structure Database (ICSD) database).

which is also present in the unheated mixture, indicating that the heating did not facilitate the formation of these species. In summary, our control experiments indicate that heating alone can polymerize P(V) above background levels, but it does not noticeably enhance reduction.

Carbon in both forms impacted the polymerization of mineral-hosted P(V) with dependence on the P(V) host (Supplementary Table S5 and Figs. S6, S7; Figs. 4, 5). The addition of OM and CB to magnesium phosphate and silica mixtures enhanced the PP(V) yield from 0.003% (without C) to 0.047% (with OM) and 0.157% (with CB), respectively, at 1150 °C. Similarly, for hydroxyapatite, OM and CB enhanced the PP(V) yield from 0.026% to 0.032% and 0.289%, respectively. These yields are low compared to that for CB + silica mixture alone, implying limited polymerization of P(V) hosted in these minerals. In contrast, adding OM and CB to vivianite and silica mixtures enhanced the polymerization yield from 0% to 1.08% and 30.98%, respectively, and produced several polymerized molecules including PP(V), PPP(V), and PPPP(V). Adding the same C sources to amorphous Fe-phosphate and silica enhanced the polymerization yield from 2.065% to 9.264% and 13.486%, respectively, and produced several polyphosphates and cyclophosphates including PP(V), PPP(V), PPPc(V), and PPPP(V). Better polymerization yield in Fe-phosphate phases points to the control of the mechanism of polyphosphate formation. XRD analysis of the experimental products revealed the presence of barringerite ($Fe_2P$), whereas no polyphosphates were detected by XRD in runs containing Fe-phosphates, despite their detection in the NaOH-EDTA extracts. This contrast suggests that in the experiments with Fe-phosphate the polyphosphates formed indirectly during partial dissolution of barringerite in the EDTA solution. This indirect mechanism of polyphosphate formation is consistent with previous studies that reported polyphosphate formation during schreibersite ($(Fe,Ni)_3P$) dissolution[26,27]. This mechanism is distinct from polyphosphate formation during control experiments and in some (Fe-free) apatite and Mg-phosphate bearing experiments, including amorphous Fe-phosphate heating at low temperatures (175–200 °C), where polyphosphate phases were likely formed directly[9], and during apatite-basalt heating where polyphosphates were formed during partial dissolution of $P_4O_{10}$[11].

Temperature seems to be an important controlling factor for P polymerization. Previous studies suggested that simple heating of metastable Fe, Ca, and Mg phosphates such as amorphous Fe-phosphate, brushite ($CaHPO_4 \cdot 2H_2O$), whitlockite ($Ca_{18}Mg_2H_2(PO_4)_{14}$), struvite

($MgNH_4PO_4 \cdot 6H_2O$), and newberyite ($MgHPO_4 \cdot 3H_2O$) can produce polymerized P(V) at lower temperatures (<300 °C)[6,10,28]. For amorphous Fe-phosphate, polymerization (generated by the indirect mechanism via barringerite dissolution) was low (0.19%) at 350 °C[9] and moderate (2.1%) at 1150 °C (this study). On the other hand, heating apatite alone did not produce polyphosphates even at 1340 °C, which is consistent with low polymerization yield at 1150 °C in the apatite + silica control experiment. Hence, although P(V) polymerization may happen (directly or indirectly) at a range of temperatures (70–1350 °C), a better yield is observed with amorphous Fe-phosphate or other metastable phosphate minerals compared to stable minerals such as vivianite and apatite at low temperatures. At higher temperatures such as 1150 °C as used in our experiments, amorphous Fe-phosphate provides a better indirect polymerization yield than apatite and vivianite. Amorphous Fe-phosphate may undergo reduction to barringerite more easily, favoring indirect polymerization during subsequent barringerite dissolution.

Carbon and temperature controlled the reduction of P(V) hosted in all four minerals (Supplementary Table S5 and Fig. S7; Figs. 4, 5). For magnesium phosphate, P(III) is the only reduced species that formed at 1150 °C upon addition of OM and CB while for hydroxyapatite, both P(I) and P(III) were formed. For magnesium phosphate, only CB produced P(III) with a yield of 0.003%, while for hydroxyapatite, both OM and CB produced P(III) and P(I) with a yield of 0.004 and 1.063%, respectively. At 1300 °C, the silica-hydroxyapatite-CB mixture produced P(I) and P(III) and gave a total reduction yield of 0.70%. Vivianite produced several reduced P species including P(I), P(III), PP(IV), and P-C(III) when OM and CB were added to the experiment at 1150 °C and the total reduction yield for these carbon sources were 2.9 and 29.28%, respectively. XRD analysis suggests the presence of $Fe_2P$ in this CB-bearing experiment (Fig. 6). At 1300 °C, the silica-vivianite-CB mixture produced similar reduced P species, including P(I), P(III), P-C(III), and PP(IV) with a total reduction yield of 17.4%. Amorphous Fe-phosphate also produced several reduced P species, including P(I), P(III), P-C(III), and PP(IV) in the presence of OM and CB at 1150 °C with the reduction yields of 8.1 and 10.2%, respectively. XRD data suggest the presence of $Fe_2P$ (Supplementary Table S7, reference XRD data are from Litasov et al.[29] and Buseck et al.[30]) in both of these amorphous Fe-phosphate experiments (Fig. 6). Previous studies suggested that $Fe^{2+}$ can reduce P(V) at moderate temperatures (200–350 °C); however, the reduction yield varied significantly in two separate studies (either <0.001%[9] or 4%[8]). Increasing temperature did enhance the reduction in one study; however, the yield was still low (0.075% at 350 °C)[9]. Even at higher temperatures explored here (e.g., 1150 °C), $Fe^{2+}$ is an ineffective P(V) reducing agent. We, therefore, suggest that C in both forms is more efficient compared to $Fe^{2+}$ in reducing P(V) in Fe-phosphate minerals, particularly at high temperatures.

Addition of an Fe source enhanced the reduction and polymerization yield of hydroxyapatite-hosted P(V) (Fig. 5, Supplementary Table S5 and Fig. S7). At 1300 °C, the silica-hydroxyapatite-CB experiment produced P(I) and P(III) with a reduction yield of 0.70%. When FeS, $Fe_3O_4$, or Fe were added to the experiment, the reduction yield increased to 81.5%, 96.2%, and 22.7%, respectively. The FeS-bearing experiment produced P(III) and several unidentified P species. XRD data indicate the presence of schreibersite ($Fe_3P$, XRD peaks are shown in Supplementary Table S7, XRD data are from previous reports[31,32]) in this experiment (Fig. 7). $Fe_3O_4$- and Fe-bearing experiments produced P(I), P(III), and P-C(III) and the latter one also produced PP(IV) as well as several polymerized P species including PP(V), PPP(V), and PPPP(V). XRD data suggest the presence of $Fe_3P$ and $Fe_2P$ in both of these two experiments (Fig. 7). $FeS_2$ addition produced a reduction yield of 0.076%; however, absolute concentrations of P(I) and P(III) were high compared to the silica-hydroxyapatite-CB experiment, and this experiment produced $Fe_3P$, which was not the case for silica-hydroxyapatite-CB. Our data thus suggest that the Fe oxidation state ($Fe^0$ for metallic Fe, $Fe^{2+}$ for FeS and $FeS_2$, and a mixture of $Fe^{2+}$ and $Fe^{3+}$ in $Fe_3O_4$) and mineralogy (oxide or sulphide) did not impact phosphide formation from apatite. We suggest that apatite can be reduced extensively in the presence of any form of Fe-oxide or -sulfide by a C source (organic

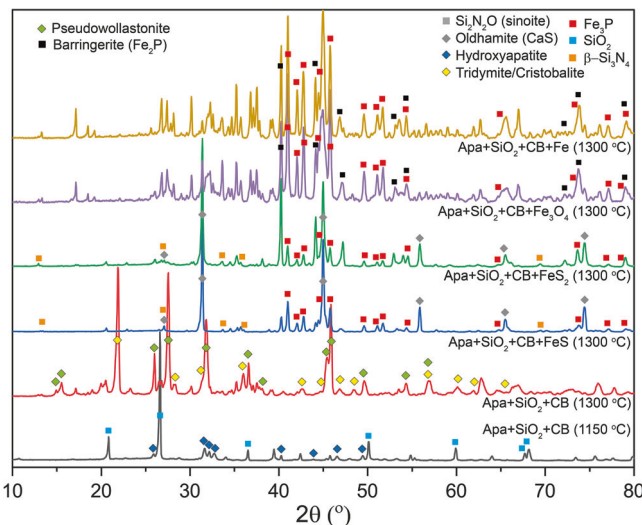

**Fig. 7 | XRD patterns of the products from hydroxyapatite experiments.**
Hydroxyapatite is not present in the Fe-bearing experiments at 1300 °C. Both
schreibersite (Fe$_3$P) and barrigerite (Fe$_2$P) or only schreibersite is present in Fe-
bearing experiments. Some phases remain unidentified (based on a comparison with
entries in the Inorganic Crystal Structure Database (ICSD) database).

matter, organic or pure carbon) in the temperature range of 1100–1300 °C
or higher.

We want to emphasize here that phosphate-host plays an important
role during P(V) reduction. It can be seen from the data above that if P(V)
is hosted in Fe-phosphates, the reduction is comparatively easier than when
P(V) is hosted in hydroxyapatite. In the case of hydroxyapatite, an addi-
tional Fe source and higher temperature were required to get similar yields
in reduction. All the experiments were conducted with hydroxyapatite, and
it remains to be explored if using fluorapatite, which is more abundant in
nature, may have any impact on the results on polymerization and reduction
yields.

**Geological evidence of magmatic P reduction and polymeriza-
tion in the Archean**

There are three possible explanations for P(III) and PP(V) observed in
Moodies Group intrusive and metasedimentary units. First, since the rela-
tive proportion of P(III) is similar in both rock units, it is possible that P(III)
abundances were reset as a consequence of regional metamorphism con-
sistent with previous experiments[8,9] and with reports of P(III) from high-
grade Eoarchean metasediments[8]. P(V) reduction in such cases required
water-poor conditions and higher temperatures, which is likely the case for
the amphibolite grade banded iron formation rocks where P(III) is pre-
viously reported[8]. However, in the Moodies Group, temperatures achieved
during greenschist-facies metamorphism (350 ± 50 °C; reaching up to
420–460 °C[20]) are much lower compared to amphibolite facies. Further-
more, it is unknown if Fe$^{2+}$-induced reduction of phosphate can happen
inside a silicate mineral containing Fe$^{2+}$, which is the case of the intrusive
samples. In case of PP(V), the magmatic units contain a much higher
concentration compared to the metasedimentary units, which cannot be
explained by regional metamorphism. We therefore discard the possibility
of metamorphic origins of P(III) and PP(V) in the Moodies intrusive and
metasedimentary units.

The second possibility is that biological processes indirectly or directly
contributed to P(III) and PP(V) observed in the metasedimentary units.
This would be consistent with the presence of widespread microbial mats in
Moodies Group[33,34]. A direct biological conversion of P(V) compounds to
P(III) or PP(V) is not known in the Archean[2]. Although P(III) may form
indirectly due to disintegration of phosphonates produced by microbial
life[35,36], genomic reconstructions suggest that microbial life did not have the

ability to generate phosphonate before the Great Oxygenation Event[37]. We
therefore discard the possibility of a direct or indirect biologic origin for the
P(III) and PP(V) in the metasediments.

The third possibility, which is most likely, is that P(III) and PP(V) in
the intrusive bodies are of magmatic origin, while in the metasedimentary
units they are recycled from the intrusive bodies (Fig. 8). Previous reports of
PP(V) occurrence in modern magmatic fumarole[11], in Phanerozoic olivine
samples from Hawaii and Pakistan[38], and in experimentally produced alkali
glass[39] supports the magmatic origin of PP(V). PP(V) in the Moodies
intrusive bodies may be hosted in glass or in olivine. Previous studies sug-
gested that weathering of basaltic rock under anoxic conditions favors the
release of P(V) into ocean water[40]. Given that P(III) and possibly PP(V) are
more soluble than P(V)[1], similar weathering as well as hydrothermal
alteration may release P(III) and PP(V) into water from the mafic igneous
rocks such as those in Moodies. We suggest that mafic rocks similar to what
we studied in the Moodies Group but of older age or from other parts of the
basin were the source of these two P species to the metasediments (Fig. 8).
Aqueous alteration of mafic rocks in the Moodies Group depositional set-
ting has been noted by previous studies[41]. In addition or alternatively, it is
also possible that the metasedimentary P(III) and PP(V) were derived from
meteorite input. Meteorites contain reduced P in the form of schreibersite
((Fe,Ni)$_3$P), and this mineral may release both P(III) and PP(V) into the
ocean during fluid-induced alteration[15,27,42]. It is conceivable that such
meteorite-derived P(III) and PP(V) existed in seawater during the deposi-
tion of the Moodies Group and thus became incorporated into the
metasediments.

**Reduction and polymerization of P during thermal metamorph-
ism and in natural conditions**

The contact zone samples from the Moodies Group suggest thermal
induced formation of polyphosphates, but not of P(V) reduction.
Experimental data from this and previous studies suggest that P(V)
polymerization is comparatively easier to do than reduction, as it may
happen even in low temperatures (<200 °C[9] vs. >1150 °C, see above) and
may be enhanced in the presence of carbon. In the Moodies Group, pyrite
formation and biotite accumulation along the contact zones, and
reworking of lithified sediment fragments into the magma, indicate that
magma emplacement postdated sedimentation and thus likely heated the
sediments at the contact zone. Previous studies also noted diverse and
widespread interactions of the mafic lava with sedimentary units at
inferred temperatures of 700–1000 °C, in the Moodies Group[16,17]. Apatite
present at the contact zones and organophosphorus compounds, the
presence of which can be speculated by the presence of organic carbon and
previous reports of microbial mats in metasedimentary units in the
Moodies Group[33], are the most plausible P(V) source in the metasediment
during heating. Although we notice pyrite at the contact zones, a close
association of apatite and pyrite is rare, if not absent. Based on these
observations, we suggest that our OM + silica, OM + hydroxyapatite, and
OM + silica + hydroxyapatite experiments (1150 °C) are the most rele-
vant for the thermal metamorphism in the Moodies Group. These
experiments produced PP(V) ranging from 0.026 to 1.81% and P(III) <
0.004%, which is consistent with the observed PP(V) (average 2.14%) in
the contact zone samples. Depending on the initial P(V) host, either
simple heating or heating in the presence of organic carbon could have
produced the observed PP(V) at the contact zones, but the conditions were
unfavorable to generate additional P(III) than existing levels. The absence
of newly generated P(III) in the contact zone samples suggest that poly-
phosphate formation in the contact zone samples did not happen via
phosphide dissolution that we have noticed in some of the experimental
samples e.g., in the viv + silica + C experiment (1150 °C). Instead, the
polyphosphates most likely formed due to PO$_4$-PO$_4$ linking in the crystal
structure of P(V)-bearing phases in the contact zones.

Our experimental data can explain other natural occurrences of P(V)
reduction. Metallic phosphides including Fe$_2$P and Fe$_3$P have been reported
from pyrometamorphosed rocks in the Hatrurim Formation (Levant

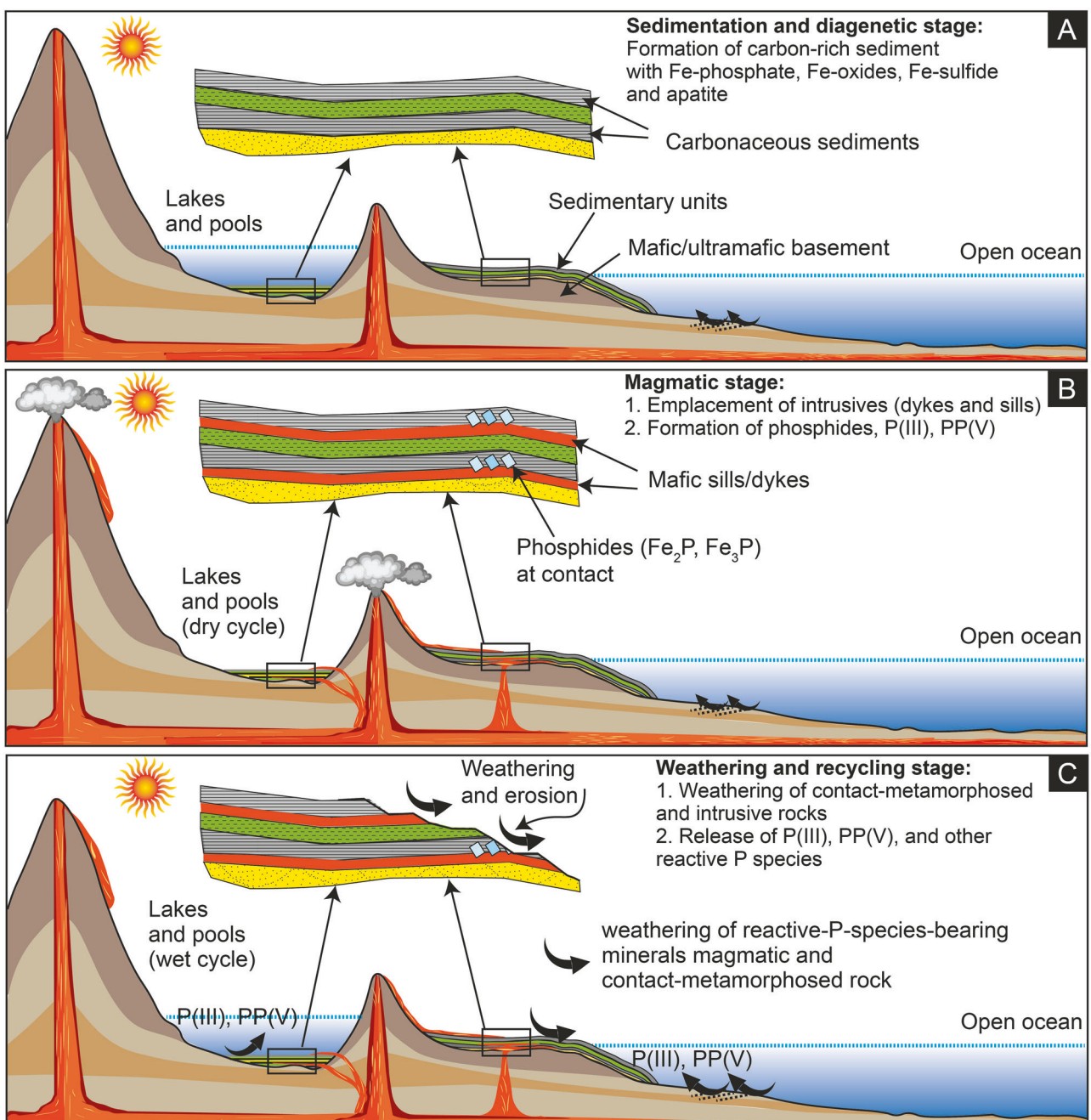

**Fig. 8 | Schematic illustrating the formation pathways and cycling of reactive phosphorus species on the early Earth. A** Sedimentary and diagenetic stage: carbon-rich sediments form in land-locked lakes or marginal marine settings, potentially containing Fe-sulfide, Fe-oxide, Fe-phosphate, and apatite. **B** Magmatic stage: volcanic intrusions (sills and dykes) penetrate the sedimentary sequence, generating phosphides, P(III), and PP(V) in heated sediments at the contact zone. PP(V) and P(III) may also occur within the intrusive and underlying basement rocks. **C** Weathering stage: erosion and weathering of contact-metamorphosed and magmatic rocks release reactive P species into lakes, pools, and the ocean, with the highest concentrations expected in closed lakes where constant supply and wet–dry cycling occur.

region) and from volcanic rocks on Disko Island[13,14,43]. In those cases, organic matter or hydrocarbons were present in association with apatite as well as Fe-minerals. The country rock was heated by magmatism at temperatures above 1050 °C in the Levant region[13] and at ca. 1200–1300 °C on Disko Island[14]. Hence, the hydroxyapatite + silica + CB + Fe-source experiments at 1300 °C broadly covers the conditions in these two places and produced similar phosphide minerals, explaining P(V) reduction in these two regions. These experiments can also explain apatite-hosted P(V) reduction in the presence of organic matter or tree-roots during lighting where temperatures reaches >1725 °C[44,45].

### Implications for the origin and evolution of early life

Our findings carry three major implications for the origin and early evolution of life on Earth. First, experimental and Moodies Group data together suggest that high-temperature alteration of sediments containing organic carbon could have been an important mechanism of polyphosphate generation in the Archean and likely on the prebiotic Earth (Fig. 8). Previous studies have shown that in the presence of organic compounds and urea-based eutectic solution with low water activity, apatite and other phosphate salt can produce polymerized species at relatively mild conditions[18,46]; however, geological evidence of such organic compounds is yet to be

discovered. In the absence of these organics, high-temperature thermally induced processes, shown in this study, could have been crucial for making polymerized P species.

Second, the Moodies Group data suggest that magmatic rocks could have been an important, long-term, and stable source of P(III) and PP(V) along with P(V) on the prebiotic Earth and in the Archean. Although the total amounts of P(III) and PP(V) in the studied basaltic rock are in the low ppm range (2.84 and 1.65 ppm), they are more water-soluble than P(V)[1], implying that anoxic weathering and hydrothermal alteration of these rocks could have liberated them more efficiently than P(V) to the ocean. Previous studies argued in favor a significant amount of P(III) along with P(V) in the Archean ocean, which has been attributed to dissolution of phosphide delivered by meteorites and produced by lightning strikes[15]. We suggest that part of the Archean seawater P(III) could have been delivered by ocean-floor weathering and hydrothermal vents (Fig. 8). The half-life of P(III) under Archean conditions has been estimated to 0.6 Myr while that of PP(V) may have much lower[6,8,47]; both of them eventually would convert into P(V). Therefore, these species could have contributed to bioavailable P(V) in the Archean ocean.

Third, although the Moodies Group represents a volcano-sedimentary succession in shallow-marine and coastal depositional settings, our experimental results are also applicable to prebiotic volcanic lakes and hot-spring environments, where thermal metamorphism or magma-sediment interaction could have taken place (Fig. 8). In these settings, organic C species may have been generated abiotically by volcanism or lightning strikes and/or delivered by meteorites[48,49] and accumulated in the lakes and pools along with phosphate minerals, metal oxides, and metal sulphides (Fig. 8A)[9,18]. Here, volcanism was common due to a significantly higher heat flux than today, driven by the combined effects of a hotter mantle and residual heat from planetary accretion[50,51], and basalt and komatiite were likely the most abundant rocks[52]. Because eruption temperatures of the latter exceeded in places 1600 °C[53], sediments in contact with komatiitic magmas would have readily achieved temperatures approaching those in our experiments (1150 and 1300 °C) or higher (Fig. 8B). Similarly, the ingredients used in the experiments, i.e., P(V) source, C, Fe-oxide/-sulphide, and heat, are also relevant to impact events and lightning strikes, which were likely common on prebiotic Earth[27,45] and known to create temperatures as high as >1700 °C[54]. We therefore suggest that there might have been several local niches on the prebiotic Earth where P(V) hosted in so-called 'unreactive' minerals such as apatite and vivianite could potentially have been transformed into phosphides, as seen in our experiments. The presence of phosphides in thermally metamorphosed rocks in the Levant region and on Disko Island[13,14] and in lightning-struck soil[44,45] further suggests the natural relevance of this reduction mechanism. Once produced, such phosphide may dissolve in water and generate the reduced and polymerized P species seen in our experimental products including P(I), P(III), P-C(IIII), PP(IV), PP(V), PPP(V), and PPPc(V), which are more reactive compared to P(V) (Fig. 8C). Because of their high solubility and reactivity, these species are more suitable phosphorylating agents for organic molecules[2,5] and thus beneficial for the origin of life. While the optimal concentrations of these species for prebiotic phosphorylation reactions remain uncertain, significant accumulations could have occurred in prebiotic pools and lakes sustained by a constant supply and subjected to wet–dry cycling—a mechanism recently suggested for P(V)[55]. Furthermore, phosphides may directly phosphorylate nucleosides and other organic compounds as well[15,56]. This reduction mechanism plausibly operated at local scale on the prebiotic Earth, particularly when the criteria for P(V) reduction were fulfilled but such local supply of reactive P species may have been sufficient to trigger phosphorylation reactions required for the origin of life.

In summary, we provide geological evidence of magmatic and thermal-metamorphic polyphosphate production and magmatic phosphite during the Archean and suggest that weathering and hydrothermal alteration of magmatic rocks can be a stable source of reactive and bioavailable P to Archean seawater. Our data suggest that P(V)-polymerization at high temperatures can be facilitated by the presence of organic carbon. Organic

carbon might be an important reducing agent at temperatures >1150 °C for P(V) hosted in so called 'unreactive' minerals such as vivianite and apatite producing phosphides, that upon dissolution can provide several soluble and reactive P species, including P(I), P(III), P-C(IIII), PP(IV), PP(V), PPP(V), and PPPc(V) crucial for prebiotic phosphorylation reactions. In conclusion, both magmatic and carbon-bearing thermally metamorphosed rock might have played important roles in supplying reactive and soluble P species during the origin and early evolution of the biosphere.

## Methods

### Whole rock analysis and TOC measurements

Powders (0.60 g each) of all samples were sent to Australian Laboratory Services (ALS) in Dublin, Ireland, for whole-rock geochemical characterisation using their method ME-MS-61r of four-acid digestion (HCl, HNO$_3$, HF, HClO$_4$) followed by ICP-MS and ICP-AES analyses. Reproducibility was assessed with rock standards OREAS-45d, OREAS-905, and MRGeo-08, and with sample replicates. For P, Fe, Cr, and Ti, the reproducibility was 5% or better. Total organic carbon (TOC) was measured at the University of St Andrews on decarbonated rock powders. Circa 0.5 g of powder were treated with 2 M HCl overnight at room temperature, followed by triple-rinsing with 18.2 MΩ·cm ultrapure water and drying in a sealed oven for three days. The dried powders were then analyzed by flash-combustion with an elemental analyzer (EA-Isolink) coupled to an IRMS (MAT253, both Thermo Fisher Scientific). Peak areas were calibrated for carbon abundances. Reproducibility is better than 5%.

### Bacteria biomass and culture conditions

Bacteria biomass (OM) used as a C source in the dry-heating experiments was generated from phosphate replete cultures of both nitrogen fixing (diazotrophic) and non-nitrogen fixing strains of cyanobacteria. The cyanobacterial diazotrophs included the freshwater *Calothrix* PCC7507 (Pasteur Culture Collection) and *Nostoc* sp., which were cultured in nitrogen-free BG11$_0$ medium[57], and the brackish dwelling strains *Nodularia spumigena* CCY 9414 (obtained from Lukas Stal, Culture Collection Yerseke, The Netherlands), and *Nodularia harveyana* SAG 44.85 (Culture Collection of Algae, Göttingen University), grown in Baltic Sea medium (Ba$_0$) lacking a combined nitrogen source[58]. Freshwater species *Microcystis aeruginosa* PCC7806 and PCC9432 were grown in nitrate containing BG11 medium[57].

An inoculum of exponentially growing culture material was used to inoculate 100 ml of the appropriate medium in T$_{175}$ ventilated cell culture flasks (Sarstedt, Germany) at a concentration of chlorophyll a of ~0.1 μg·ml$^{-1}$ and incubated at 24 °C, 60 μmol photons m$^{-2}$ s$^{-1}$ on a 14:10 hour day-night cycle, under present-day atmospheric conditions (Plant growth chamber E-22L, Percival, USA). Biomass was harvested after 4–6 weeks once the cultures had reached stationary phase[57,58]. Cultures were pelleted in sterile 50 ml polypropylene tubes (Sarstedt, Germany) by centrifugation at 7500 rcf for 30 min, washed twice with sterile MQ water and frozen at –80 °C. The pellets were lyophilized (at –10 °C, 0.04 mbar; CHRIST LSC plus) and the dried biomass powdered using an agate mortar and pestle. The final mixture of biomass (referred to as OM in the main text) includes 0.26 g *Calothrix* PCC7507, 0.17 g *N. harveyana*, 0.19 g *Nostoc*, 0.18 g *N. spumigena*, 0.70 g *M. aeruginosa* PCC7806, and 0.67 g *M. aeruginosa* PCC9432. While these particular diazotrophs are not representative of deep-branching cyanobacteria, their OM was not found to significantly vary with respect to C and N contents, nor stable isotope signatures, when grown under anoxic conditions simulating those on early Earth, present-day atmospheric conditions, nor under elevated atmospheric CO$_2$ conditions[58]. The OM thus used in these experiments can be considered as an Archean equivalent of organic-carbon-containing matter.

### Heating experiments

All experiments and associated analyses were carried out at the University of St Andrews. Acid-washed (1 M or 2 M HCl) and baked (500 °C) glass containers and acid- and hot-water (18.2 MΩ·cm, ultrapure) washed plastic centrifuge tubes, bottles, pipette tips, and syringes were used in all the stages

https://doi.org/10.1038/s43247-025-02824-x **Article**

of the experiments and subsequent sampling. Synthetic hydroxyapatite ($Ca_5(PO_4)_3(OH)$; Thermo Scientific; Cat. No. 036731.36, Lot. X15F024) and magnesium phosphate ($Mg_3(PO_4)_2 \cdot xH_2O$; Sigma-Aldrich; PCode 1002982444), natural vivianite (Brazil), and in-house prepared amorphous Fe-phosphate were used as a P(V) precursor. To prepare amorphous Fe-phosphate, $FeCl_2 \cdot 4H_2O$ (Sigma Aldrich, PCode 101074277) and $(NH_4)H_2PO_4$, (Thermo Scientific; Cat No. 193701000, Lot A0443028) were added in a molar ratio of 3:1 in 300 ml deoxygenated, deionized water (18.2 MΩ·cm) with a pH of 4 (this method is adopted from Herschy et al.[8] and Baidya et al.[9]). The glass bottle was connected to a vacuum pump and $N_2$ cylinder for maintaining an anoxic condition and a hot plate for increasing temperature. The solution was stirred in the dark (shielded with Al foil), evaporated into dryness maintaining anoxic conditions at 60 °C, and the solid residue was brought back to room temperature. The residue was crushed with pestle & mortar, stored in a sealed vial, and used as a P(V) precursor for subsequent heating experiments. OM as described above and carbon black from Cabot were used as a C source. Metallic Fe (Thermo Scientific, Cat. No. 000737.30, Lot. R20F039), $Fe_3O_4$ (Inoxia Limited (UK); EC 215-169-8), FeS (Thermo Scientific; Cat No. 014024.09, Lot. T07H028) and $FeS_2$ (natural; Thermo Scientific; Cat. No. 042633.06, Lot. T23G054) were used as an Fe-source in selected apatite-bearing experiments. Additionally, $SiO_2$ powder (crushed Sea sands, VWR, CAS 14808-60-7) was used to prepare the initial mixture.

For each experiment, 0.6 g of powdered mixture was prepared by mixing 0.18 g of one of the phosphate sources, 0.18 g of one of the C sources, variable proportions of powdered $SiO_2$, and in some cases, a Fe source using a pestle & mortar (Supplementary Table 1). The weight ratio for C and P(V) was 1:1 except for the control experiments. Variable weights of Fe precursors were used to maintain the total Fe added in each experiment the same. As an example, in Exp 8, 0.18 g of hydroxyapatite, 0.18 g of OM, and 0.24 g of silica powder were mixed. For control experiments containing only $SiO_2$ powder, 0.6 g od $SiO_2$ powder was used while those containing $SiO_2$ powder and a P(V) source or silica powder and a C source, 0.18 g of C or P(V) source was mixed with 0.42 g of $SiO_2$. 0.52–0.56 g of the powder mixtures were pressed into pellets, each pellet containing 0.13–0.14 g of powder mixture.

The pellets were then loaded into an alumina boat and kept inside a tube furnace under flowing $N_2$ gas (30 ml/min) at room temperature for 3 h. This step was done to make sure the atmosphere inside the tube is inert before the heating stage started. The furnace was equipped with a K-type thermocouple and a temperature controller. A forward heating ramp of 7-8 °C/min was used to reach the desired temperatures, i.e., 1150 °C or 1300 °C, which were maintained for 48 h, followed by a downward cooling ramp of 5 °C/min (Supplementary Fig. S1B). The heated residues were brought back to room temperature while maintaining the same $N_2$ flow rate. They were then powdered and stored in sealed vials before subsequent analysis.

### Solid characterization using powder X-ray diffraction (PXRD)
An aliquot of the powdered experimental products was loaded into 0.5- or 0.7-mm capillary tubes and sealed for XRD analysis. The PXRD patterns were recorded on a STOE STADIP diffractometer using Mo Kα1 radiation at room temperature, over a 2θ range of 2.5° to 40°, with a step size of 2.75° per step and acquisition times of 1 h 20 min to 1 h 25 min in capillary Debye-Scherrer mode. For consistency and ease of comparison with standard reference patterns, we converted the Mo Kα data to Cu Kα equivalents using the PowDLL software. The 2θ values presented in the plots (Figs. 6 and 7) therefore, reflect Cu Kα wavelengths. The PXRD data were compared to solids in the Inorganic Crystal Structure Database (ICSD) and Crystallography Open Database (COD) for phase identification using the Crystal Diffract software (version 7.0.4.300). Raw data are provided in Supplementary Data S2.

### NMR and IC-ICPMS analysis for P speciation
The P speciation analysis for the rock samples were done using an ion chromatography-inductively coupled plasma mass spectrometer (IC-

ICPMS) set-up at the University of St Andrews[19]. For the experimental samples, both NMR spectroscopy and IC-ICPMS were used. It is important to note that NMR has higher detection limits (150–200 ppb) for P species in the used analytical conditions while the IC-ICPMS set-up can detect as little as 0.1 ppb of P species in solution. Furthermore, NMR can detect several P species while only inorganic P(I), P(III), P(V), and PP(V) can be detected in the IC-ICPMS set-up.

For each rock sample, 0.20 g powder was treated with an Ethylenediaminetetraacetic acid-sodium hydroxide (0.05 M EDTA and 0.25 M NaOH) solution[59] maintaining a solid:solution ratio of 1:10, 1:5, or 1:3.75. As the total concentration of P species, particularly P(III) and PP(V), in the rock samples were low, three different solid:solution ratios were chosen. For preparing the EDTA-NaOH solution, $Na_2EDTA$ (Sigma Aldrich) salt and 10 M NaOH solution (Thermo Scientific) were dissolved in deionized water. The solid and solution mixture were prepared in 15 ml Falcon tubes (acid- and hot-water washed) and left on a shaker (175 rpm/min) for 14–15 h. The solid-solution mixtures were then centrifuged at 3000 rpm for 15 min. For the metasedimentary and contact zone samples, a floating grey phase was observed in the supernatant, which most likely is organic carbon. For these samples, the supernatant was filtered by 0.20 μm PTFE syringe filter (Fisherbrand). A transparent solution at this stage suggests the precipitation of all the extracted Fe and other transition metals. Precipitation of these metals is essential for the P speciation measurements using the subsequent IC-ICPMS analysis[19] because excess dissolved metal may precipitate as oxides in the anion separation column of the IC and bind phosphate by adsorption within the column, thereby impacting analytical quality. The supernatant was then diluted 25 or 50 times and P species analysis was done using the IC-ICPMS set-up.

For each experimental sample, 0.1 g of powder was treated with 1 ml of EDTA-NaOH solution for 8 h. A 4 ml, flat-bottom glass vial with a magnetic stirrer was used for this step. After this, the supernatant was filtered with 0.20 μM PTFE syringe filter (Fisher brand). The solution was then diluted 25, 50, or 75 times and analyzed with IC-ICP-MS. An aliquot of the solution was analysed with an NMR.

The IC-ICPMS set-up consisted of a Dionex ICS-6000 ion chromatograph (IC) and an Element 2 inductively coupled plasma mass spectrometer (ICP-MS) (both from Thermo Scientific). The IC was equipped with a Dionex AS-AP autosampler, a 25 mm Dionex IonPac AS17-C separation column (2 mm bore), a 25 mm Dionex IonPac AG17-G guard column (2 mm bore), and a Dionex ADRS-600 (2 mm) suppressor. The flow rate in the IC was held constant at 0.5 ml/min. The concentration of the KOH eluent solution was ramped up from 1 mM to 40 mM over 20 min, which was then maintained for another 24 min followed by a ramp-down to 1 mM in 6 min. The detector outlet of the IC was physically connected to a 1 ml/min nebulizer attached to the spray chamber (Scott model; quartz glass) of the ICP-MS. The ICP-MS was operated at a sample gas flow rate of 1.03–1.10 ml/min, cool gas flowrate of 16 ml/min, and RF power of 1183 in medium-resolution mode. Data were collected in the ICP-MS as chromatographs of 3 min duration. Each chromatogram consisted of the pre-peak background, the peak, and the post-peak background for each P-species.

The chromatographic data were produced for samples and standards. Standards contained 0.2 to 100 ppb of four P species (prepared from $NaH_2PO_2 \cdot H_2O$ (Thermo Fisher), $Na_2HPO_3 \cdot 5H_2O$ (Thermo Fisher), $NaH_2PO_4$ (Thermo Fisher), and $Na_4P_2O_7$ (Sigma Aldrich)) in a solution matrix similar to the samples. The acquired chromatogram data were smoothened with the OriginLab software, using the FFT filter with a points-of-window value of 5, and the peak area under the curve was used for quantification. The detection limits of the IC-ICPMS set-up were <0.1 ppb for P(III) and P(V), 0.1 ppb for P(I), and 0.2 ppb for PP(V). Analysis was done two or three times, and the geometric mean was calculated due to large variations in the replicates.

For NMR analyses, an aliquot (540 μL) of EDTA-NaOH extracts of the experimental samples was mixed with 60 μL of $D_2O$ and loaded in an NMR tube to be measured on a Bruker AVIII 500 MHz NMR. This NMR set-up is equipped with a nitrogen-cooled broadband cryoprobe, which improves

sensitivity. Samples were analyzed in proton-decoupled mode with 15,000 scans. The $^{29}$P chemical shifts were referenced to phosphoric acid, which has a chemical shift of 0δ. Standards containing six different P species, P-C(III), P(I), P(III), P(V), PP(V), PPP(V), PPPc(V) of known concentrations (0.2 ppm to 800 ppm phosphorus), prepared using $NaH_2PO_2.H_2O$ (Thermo Scientific), $Na_2HPO_3·5H_2O$ (Thermo Scientific), $NaH_2PO_4$ (Thermo Scientific), and $Na_4P_2O_7$ (Sigma Aldrich), $Na_5P_3O_{10}$,(Thermo Scientific), and $Na_5P_3O_9$ (Thermo Scientific) were analysed to build calibration curves for each P species. The experimental samples contained PPPP(V) or even longer-chain polyphosphates as well as several unknown P species. For PPPP(V), the calibration curve of PPP(V) was used; for the unknown P species, the calibration of P(V). The raw data of the NMR analyses are provided as Supplementary Data SD2.

## Data availability

All data are available in the main text or in the supplementary materials. The data are also available through the National Geoscience Data Centre, hosted by the British Geological Survey (BGS), at https://doi.org/10.5285/fe259ea3-256e-436d-81a6-9be3f63d2476

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

## Acknowledgements

This work was financially supported by a Natural Environment Research Council (NERC- UKRI) Frontiers grant to EES (NE/V010824/1), a Marie Skłodowska-Curie Actions grant to ASB (EP/Y026497/1), and a German Research Foundation (DFG) grant (GE2558/4-1) to MMG. We acknowledge Sami Mikhail for his help during pellet making and Tomas Lebl and Siobhan Smith for their help during the NMR analysis. Rajan Biswas and Oxana Magdysyuk helped during XRD analyses and data interpretation. For XRD analyses, we also acknowledge the Engineering and Physical Science Research Council (EPSRC) Core Equipment Grant (EP/V034138/1). ICDP staff provided access to core and administration of the sample database. Sebastian Reimann drafted the geologic map. South Africa's Council for Geoscience supplied the hyperspectral core maps shown in Figure. S1. The complete data for this study are available through the National Geoscience Data Centre of the British Geological Survey. In order to meet institutional and research funder open-access requirements, any accepted manuscript arising shall be open access under a Creative Commons Attribution (CC BY) reuse licence with zero embargo.

## Author contributions

The idea is conceived by E.E.S. and A.S.B. M.M.G. and C.H. supplied bacteria extracts and rock samples, respectively. A.S.B. analyzed the samples and did the experimental investigations with the help of E.E.S. and C.S. A.S.B. prepared the first draft of the manuscript with contribution from M.M.G. and C.H. The manuscript was reviewed and edited by E.E.S., A.S.B., C.H. and M.M.G.

## Competing interests

The authors declare no competing interests.
