## [Transparent Peer Review file · Communications Earth & Environment]

Magmatic and thermally produced reactive phosphorus 3.2 billion years ago and its implications for early life

Corresponding Author: Dr Abu Baidya

Version 0:

Decision Letter:

Dear Dr Baidya,

Your manuscript titled "Magmatic and thermally produced reactive phosphorus 3.2 billion years ago: Implications for early life" has now been seen by 2 reviewers, and we include their comments at the end of this message. They find your work of interest, but some important points are raised. We are interested in the possibility of publishing your study in Communications Earth & Environment, but would like to consider your responses to these concerns and assess a revised manuscript before we make a final decision on publication.

We therefore invite you to revise and resubmit your manuscript, along with a point-by-point response that takes into account the points raised. Please highlight all changes in the manuscript text file.

Please submit your point-by-point responses as a separate file, distinct from your cover letter where you can add responses to the Editors' comments that you do not want to be made available to the reviewers. Word files are preferred. We recommend that any figures, tables or graphs that are included in the response to reviewers are also included in the main article or Supplementary Information.

Please use the following link to submit your revised manuscript, point-by-point response to the referees' comments (which should be in a separate document to any cover letter), a tracked-changes version of the manuscript (as a PDF file) and the completed checklist:

Link Redacted

We hope to receive your revised paper within six weeks; please let us know if you aren't able to submit it within this time so that we can discuss how best to proceed. If we don't hear from you, and the revision process takes significantly longer, we may close your file. In this event, we will still be happy to reconsider your paper at a later date, as long as nothing similar has been accepted for publication at Communications Earth & Environment or published elsewhere in the meantime.

Please do not hesitate to contact us if you have any questions or would like to discuss these revisions further. We look forward to seeing the revised manuscript and thank you for the opportunity to review your work.

Best regards,

Alireza Bahadori, PhD
Associate Editor
Communications Earth & Environment

EDITORIAL POLICIES AND FORMATTING

- Behavioural and social science
- Ecological, evolutionary & environmental sciences
- Life sciences

Furthermore, please align your manuscript with our format requirements, which are summarized on the following checklist: <https://www.nature.com/documents/commsj-phys-style-formatting-checklist-article.pdf> Communications Earth & Environment formatting checklist

and also in our style and formatting guide <https://www.nature.com/documents/commsj-phys-style-formatting-guide-accept.pdf> Communications Earth & Environment formatting guide .

*** DATA: Communications Earth & Environment endorses the principles of the Enabling FAIR data project (<http://www.copdess.org/enabling-fair-data-project/>). We ask authors to make the data that support their conclusions available in permanent, publically accessible data repositories. (Please contact the editor if you are unable to make your data available).

All Communications Earth & Environment manuscripts must include a section titled "Data Availability" at the end of the Methods section or main text (if no Methods). More information on this policy, is available at <http://www.nature.com/authors/policies/data/data-availability-statements-data-citations.pdf> <http://www.nature.com/authors/policies/data/data-availability-statements-data-citations.pdf>.

DATA SOURCES: All new data associated with the paper should be placed in a persistent repository where they can be freely and enduringly accessed. We recommend submitting the data to discipline-specific, community-recognized repositories, where possible and a list of recommended repositories is provided at <http://www.nature.com/sdata/policies/repositories> <http://www.nature.com/sdata/policies/repositories>.

If a community resource is unavailable, data can be submitted to generalist repositories such as <https://figshare.com/> Figshare or <http://datadryad.org/> Dryad Digital Repository. Please provide a unique identifier for the data (for example a DOI or a permanent URL) in the data availability statement, if possible. If the repository does not provide identifiers, we encourage authors to supply the search terms that will return the data. For data that have been obtained from publically available sources, please provide a URL and the specific data product name in the data availability statement. Data with a DOI should be further cited in the methods reference section.

Please refer to our data policies at <http://www.nature.com/authors/policies/availability.html> <http://www.nature.com/authors/policies/availability.html>.

REVIEWER COMMENTS:

Reviewer #1 (Remarks to the Author):

Baidya et al conducted an analytical and experimental campaign to understand whether sources of reduced phosphorus produced via endogenous high-temperature processes would have been available on the early Earth. The paper presents total P and P speciation measurements for the Moodie supergroup, an Archean succession featuring basaltic contact metamorphism with C-rich sediments. Supplementing these results, the paper presents the findings of anoxic high-T experiments that probe yields and speciation of reduced P in various analogue experimental set-ups.

I am impressed with the paper. The approach is thorough, with no glaring oversimplifications/errors. The findings seem

robust, with the experimental yields roughly aligning with the real geochemical data. All in all, this is quite convincing.

I do have some questions for the authors that I would appreciate seeing some edits to address.

1) If contact metamorphism is responsible for most of the reduced P in the Moodie rocks studied, why is there no apparent trend in reduced P % away from the contact zone? I couldn't find a clear explanation of this in the paper. There is a section of text suggesting transport of reduced P out of the contact zone into the sediments. However, such a process would presumably produce a gradient in reduced P % - a mixing ratio – and we do seem to see a mixing ratio of P contents in the bulk chemistry data (Fig. 3). The presence of this mixing ratio for P content but not P speciation is perhaps the most puzzling aspect of the data in my view. Please address this in some detail.

2) I did not follow this statement: "Barringerite (Fe₂P) but not any polyphosphates are detected in experiments containing Fe-phosphates, suggesting the formation of polyphosphates during dissolution in the EDTA solution." Why does this follow? What am I supposed to take from this? If polyphosphates are only forming when samples are exposed to EDTA, doesn't that compromise the study?

3) I find this part of the discussion a bit reaching: "We suggest that part of the Archean seawater P(III) could have been delivered by ocean-floor weathering and hydrothermal vents. The half-life of P(III) under Archean conditions has been estimated to 0.6 Ma while that of PP(V) may have much lower 6,8,42 ; both of them eventually would convert into P(V). Therefore, these species could have contributed to bioavailable P(V) in the Archean ocean." The study is of crustal contact metamorphism, but the statement above focusses on the ocean. Is the argument that contact metamorphism is happening in the feeding zones of hydrothermal vents? It just seems odd as no relevant samples were studied and there is no modelling on this topic in the paper. On the other hand, weathering of reduced P from relevant rocks on land seems very likely, as does reduced P production in most heating environments involving P, org-C, and iron, of which we can imagine many on land.

4) Could more thought be put into the best way to get the reduced P involved in prebiotic chemistry? It's publication worthy to know that there are sources, but the concentrations involved are low. What, in the authors view, is the absolute best way to get the high concentration of reduced P via these sources?

5) Panel D in Fig. 3 seems to have the wrong colour scheme? Some of the points are black, rather than green.

I recommend minor revisions.

Reviewer #2 (Remarks to the Author):

See the attachment.

** Visit Nature Portfolio's author and referees' website at www.nature.com/authors for information about policies, services and author benefits**

Communications Earth & Environment is committed to improving transparency in authorship. As part of our efforts in this direction, we are now requesting that all authors identified as 'corresponding author' create and link their Open Researcher and Contributor Identifier (ORCID) with their account on the Manuscript Tracking System prior to acceptance. ORCID helps the scientific community achieve unambiguous attribution of all scholarly contributions. You can create and link your ORCID from the home page of the Manuscript Tracking System by clicking on 'Modify my Springer Nature account' and following the instructions in the link below. Please also inform all co-authors that they can add their ORCIDs to their accounts and that they must do so prior to acceptance.

If you experience problems in linking your ORCID, please contact the Platform Support Helpdesk.

Version 1:

Decision Letter:

Dear Dr Baidya,

Your revised manuscript titled "Magmatic and thermally produced reactive phosphorus 3.2 billion years ago: Implications for early life" has now been seen by our reviewers, whose comments appear below. In light of their advice we are delighted to say that we are happy, in principle, to publish a suitably revised version in Communications Earth & Environment.

We therefore invite you to revise your paper one last time to address the remaining concerns of our reviewer 2. At the same time we ask that you edit your manuscript to comply with our format requirements and to maximise the accessibility and therefore the impact of your work.

EDITORIAL REQUESTS:

*****Please take care to match our formatting and policy requirements. We will check revised manuscript and return manuscripts that do not comply. Such requests will lead to delays. *****

SUBMISSION INFORMATION:

OPEN ACCESS:

Communications Earth & Environment is a fully open access journal. Articles are made freely accessible on publication. For further information about article processing charges, open access funding, and advice and support from Nature Portfolio, please visit <https://www.nature.com/commsenv/open-access>

Link Redacted

Best regards,

Alireza Bahadori, PhD
Senior Editor
Communications Earth & Environment
Consulting Editor
Communications Sustainability

REVIEWERS' COMMENTS:

Reviewer #1 (Remarks to the Author):

Everything is addressed to my satisfaction.

Reviewer #2 (Remarks to the Author):

Dear Authors,

Thank you for all the effort and time put into addressing the reviewers' comments. I believe all of my comments have been addressed to a satisfactory degree.

I also read the other reviews and your comments and responses to those, and from what I can ascertain most of these have been addressed as well, leading to a much improved product.

Line 478: remove one "over a 2θ range of 2.5° 40°"

Kindly regards,
Jérémie Aubineau

** Visit Nature Portfolio's author and referees' website at www.nature.com/authors for information about policies, services and author benefits**

We thank both reviewers for their careful evaluation of our manuscript and for providing constructive feedback. Below, we present our point-by-point responses to their comments (in blue text). The revised manuscript has been updated accordingly based on these responses. In the revised manuscript, edits are marked with blue text.

Responses to Reviewer 1:

Baidya et al. conducted an analytical and experimental campaign to understand whether sources of reduced phosphorus produced via endogenous high-temperature processes would have been available on the early Earth. The paper presents total P and P speciation measurements for the Moodies Group, an Archean succession featuring basaltic contact metamorphism with C-rich sediments. Supplementing these results, the paper presents the findings of anoxic high-T experiments that probe yields and speciation of reduced P in various analogue experimental set-ups.

I am impressed with the paper. The approach is thorough, with no glaring oversimplifications/errors. The findings seem robust, with the experimental yields roughly aligning with the real geochemical data. All in all, this is quite convincing.

Response: We thank the reviewer for the positive comments.

I do have some questions for the authors that I would appreciate seeing some edits to address.

1) If contact metamorphism is responsible for most of the reduced P in the Moodies rocks studied, why is there no apparent trend in reduced P % away from the contact zone? I couldn't find a clear explanation of this in the paper. There is a section of text suggesting transport of reduced P out of the contact zone into the sediments. However, such a process would presumably produce a gradient in reduced P % - a mixing ratio – and we do seem to see a mixing ratio of P contents in the bulk chemistry data (Fig. 3). The presence of this mixing ratio for P content but not P speciation is perhaps the most puzzling aspect of the data in my view. Please address this in some detail.

Response: We thank the reviewer for their thoughtful comments and the opportunity to clarify our interpretations.

First, we would like to clarify that we do not attribute the formation of reduced phosphorus species (P(III)) in the Moodies Group to contact metamorphism. In our high-temperature experiments (≥ 1150 °C), we observed that carbon can act as a reducing agent for phosphate, particularly when hosted in Fe-phosphates or hydroxyapatite (in the presence of Fe-phases). However, this process was not observed in the Moodies Group samples. Based on our mineralogical and textural observations, we consider our OM + silica, OM + hydroxyapatite, and OM + silica + hydroxyapatite experiments at 1150 °C to be the most relevant analogues for thermal metamorphism in the Moodies context. These specific experiments yielded negligible amounts of P(III) (yield $< 0.004\%$), which is consistent with the very limited enrichment of P(III) observed in the contact zone between the intrusive rocks and the sediments. We emphasize this point in line 312-330 in the revised manuscript.

We suggest that P(III) observed in the intrusive rocks are primary and magmatic in origin, and that the reduced phosphorus species found in the adjacent sedimentary rocks likely originated through subsequent weathering or hydrothermal alteration of these igneous intrusions. Importantly, we do not suggest that the specific intrusive samples analyzed in this study have been altered and subsequently delivered P(III) to the surrounding sediments. If that had been

the case, we would expect to observe systematic trends in the P(III) chemistry that are not present. Instead, we point to the possibility that the P(III) in the sediments may be derived from other sources, such as erosion of older mafic rocks. It is also possible that other more altered portions of the intrusion released P(III) into the environment that was subsequently trapped in the sediments. Such alteration has, for example, been noted by Stengel et al. (2024, *South African Journal of Geology*). Additionally, we acknowledge the potential role of meteoritic input as a supplementary source of reduced phosphorus in the sedimentary rocks, as discussed by Pasek et al. (2013, *PNAS*). These are explained in lines 294-309 in the revised manuscript.

Regarding the mixing line in Fig. 3, we would like to clarify that it refers to physical mixing of materials derived from the intrusive rocks and the sedimentary units. This approach is appropriate, because cut samples included varying proportions of both mafic igneous and sedimentary material (which was likely unconsolidated) (e.g., see Fig. S4). The mixing ratios are a way of quantifying those proportions. While the total phosphorus concentrations differ substantially between the intrusive (590–610 ppm) and sedimentary rocks (310–360 ppm), the estimated concentrations of P(III) are broadly similar (0.43–2.84 ppm and 0.28–1.64 ppm, respectively). Due to the significant difference in total phosphorus content, the contact zone exhibits intermediate P concentrations, reflecting a mixture of these two lithologies. In contrast, the similarity in P(III) concentrations across both rock types results in a trend line that is nearly horizontal with respect to the x-axis, indicating little variation in P(III) abundance as a function of lithology.

In the case of PP(V), the contact samples deviate from the mixing line, because additional PP(V) was generated at the contact, and hence the abundance of PP(V) in the contact samples cannot be explained by mixing of igneous and sedimentary materials alone. We have revised the text to emphasize this point in the revised manuscript (line 169-176).

2) I did not follow this statement: “Barringerite (Fe₂P) but not any polyphosphates are detected in experiments containing Fe-phosphates, suggesting the formation of polyphosphates during dissolution in the EDTA solution.” Why does this follow? What am I supposed to take from this? If polyphosphates are only forming when samples are exposed to EDTA, doesn't that compromise the study?

Response: Thank you for pointing this out. We agree that our initial explanation of polyphosphate formation may not have been sufficiently clear.

Polyphosphate formation in our experimental samples can occur via two distinct mechanisms. First, carbon can directly facilitate the formation of polyphosphates, as observed in the silica + carbon black (CB) experiment, which produced detectable amounts of pyrophosphate. Second, polyphosphates may also form during the post-experimental dissolution of phosphide phases. Specifically, experimental products containing barringerite and/or schreibersite can yield polyphosphates when treated with EDTA-NaOH during the extraction process.

Among these, the first mechanism—carbon-facilitated polyphosphate formation—is more relevant to the conditions represented by the Moodies samples. We have clarified this distinction and its implications in the revised manuscript text (see line 212-218 and 330-334).

3) I find this part of the discussion a bit reaching: “We suggest that part of the Archean seawater P(III) could have been delivered by ocean-floor weathering and hydrothermal vents. The half-life of P(III) under Archean conditions has been estimated to 0.6 Ma while that of PP(V) may have much lower 6,8,42 ; both of them eventually would convert into P(V). Therefore, these species could have contributed to bioavailable P(V) in the Archean ocean.”

The study is of crustal contact metamorphism, but the statement above focusses on the ocean. Is the argument that contact metamorphism is happening in the feeding zones of hydrothermal vents? It just seems odd as no relevant samples were studied and there is no modelling on this topic in the paper. On the other hand, weathering of reduced P from relevant rocks on land seems very likely, as does reduced P production in most heating environments involving P, org-C, and iron, of which we can imagine many on land.

Response: We thank the reviewer for this insightful comment. In our study, we identify two key mechanisms that contribute to the formation of reduced and polymerized phosphorus species.

The first is contact metamorphism, as correctly noted by the reviewer. We agree that, under favourable physicochemical conditions, contact metamorphism can generate both reduced (P(III)) and polymerized phosphorus (PP(V)) species. These species can subsequently be mobilized through weathering processes and transported to the ocean, as the reviewer suggests.

In addition, our results indicate that intrusive basaltic rocks themselves may host both reduced and polymerized phosphorus species of likely magmatic origin. This implies that basaltic rocks on the ocean floor may similarly contain P(III) and PP(V). Previous studies have proposed that weathering of basaltic crust in the Archean could have acted as a source of phosphate to the oceans (e.g., Syverson et al., 2022, *Geophysical Research Letters*). Building on this, we suggest that reduced and polymerized phosphorus species derived from oceanic basalt could also contribute to the phosphorus inventory of the early ocean. These species would eventually oxidize to phosphate (P(V)), thereby enhancing the pool of bioavailable phosphorus in the Archean ocean.

We have revised the manuscript to better emphasize this dual mechanism and its implications for early ocean chemistry. A more advance computational model of the weathering fluxes from igneous rocks is beyond the scope of this study, but we are in the process of investigating this point further.

4) Could more thought be put into the best way to get the reduced P involved in prebiotic chemistry? It's publication worthy to know that there are sources, but the concentrations involved are low. What, in the authors view, is the absolute best way to get the high concentration of reduced P via these sources?

Response: Recent work by Walton et al. (2025, *Science Advances*) showed that high concentrations of P(V) could have accumulated in prebiotic volcanic lakes due to evapo-concentration after weathering, and we propose that a similar mechanism could have operated to also concentrate reactive P species. We have added this point to the manuscript (line 385-387). A volcanic evaporitic setting, where intrusions generated reactive P in the surrounding environment, would have been ideal for our proposed mechanism.

5) Panel D in Fig. 3 seems to have the wrong colour scheme? Some of the points are black, rather than green.

Response: Thank you for pointing this out. We corrected it.

I recommend minor revisions.

Responses to Reviewer 2

Summary: The authors present an interesting study in which they performed mineralogical and geochemical analyses of Archean metasediments combined with laboratory experiments mimicking thermal metamorphism of organic carbon- and phosphorus-bearing sediments. Supported by their experimental works, the authors reveal the likely origin of polyphosphates and reduced phosphorus species that are more soluble and reactive than phosphate ions. These findings bring new insights for deciphering the origin of life in the prebiotic world of the early Earth.

I really appreciated the presented manuscript. Overall, the manuscript is well written, interesting and relatively easy to understand. The introduction and discussion sections are clear, and I did not find any major flaws in the data interpretation. However, I have major/moderate comments regarding the results and XRD methodology. They need to be addressed before publication. I hope this will not change the data interpretation.

The methodology seems appropriate to the conducted study, although I do not have any specialist knowledge on heating experiments and nuclear magnetic resonance spectroscopy. Therefore, I can only offer limited judgement in this regard. This research offers a novel viewpoint by exploring polyphosphates and reduced phosphorus species in rocks from the early Earth, and it should be of interest to readers of *Communication Earth & Environment*. As I am not a native English speaker, I would not feel comfortable taking responsibility for a full linguistic review.

Response: We thank the reviewer for the positive comments.

Major/moderate comments:

(1) I do not understand why the authors used the geometric mean. There is no mathematical explanation. The geometric mean is usually used for calculating the mean of a consecutive percentage of when the data follow a lognormal distribution. Use 2 standard deviations around the arithmetic mean for representing the variability of the data. ISO 5725 uses two terms “trueness” and “precision” to describe the accuracy of a measurement method. “Trueness” refers to the closeness of agreement between the arithmetic mean of a large number of test results and the true or accepted reference value. “**Precision**” refers to the closeness of agreement between test results (ISO 5725-1, 1994). Measurement precision is used to define measurement repeatability. Also, if you want to keep the geometric mean, please clearly explain, and you have to calculate the **geometric** standard deviation.

Response: The geometric mean is appropriate for datasets that include ratios and where the data show relatively wide scatter. The arithmetic mean would be skewed towards the higher values, which this approach tends to avoid. It is thus the more conservative method, but we think it captures more accurately the central tendencies of our dataset. The standard deviations shown in the figure are also geometric errors. We have clarified this point in the caption of Fig. 3.

In figure 3, I recommend changing Fig 3E-H to A-D for consistency with the text.

Response: We have corrected this in the revised manuscript.

Clarify in the caption that the data are presented in Table S4 because I thought the data were wrong as I looked at the table S3.

Response: We have mentioned this in the Figure caption.

And finally, why are the data plotted as a function of Cr/Ti? This needs to be clarified.

Response: We thank the reviewer for this helpful observation. We selected the Cr/Ti ratio for the x-axis because it effectively distinguishes between the intrusive and sedimentary rock units, providing a clear compositional contrast. Both Cr and Ti are fluid-immobile elements (Ptáček et al. 2020 EPSL) and therefore suitable for this exercise. This makes it a useful proxy for identifying physical mixing relationships in the contact zone samples.

The contact zone samples represent mixtures of the two lithological end-members, because during the sample cutting some proportions of sediments and some proportions of igneous material were included on either side of the contact. Because the Cr/Ti ratios are so distinct between the two lithologies, they can be used to quantify their relative proportions. That is why the Cr/Ti values of the contact zone samples fall between those of the intrusive and sedimentary rocks. If the concentration of a particular element in the contact zone lies along this mixing line, it suggests that the element's abundance can be explained by simple physical mixing—for example, as observed for total phosphorus.

In contrast, if a species plots significantly off the mixing line, it implies that its presence is not due solely to physical mixing, but may involve additional processes. For instance, in the case of the Moodies Group, the enrichment in PP(V) in the contact zone is interpreted to result from thermal metamorphism at the contact rather than mixing alone. We have clarified this interpretation in the revised manuscript text (caption of Fig. 3).

(2) Apatite. In the Archean metasediments, you show the presence of apatite minerals. Are they fluorapatite or hydroxyapatite? Fluorapatite is a more common minerals in magmatic rocks. Isn't it? This is not clear in the SEM images; you could realize some EDS measurements to confirm the presence of fluorapatite or hydroxyapatite. Because I am wondering whether the experiments would have provided similar results with fluorapatite instead of hydroxyapatite.

Response: We thank the reviewer for this observation. We indeed conducted EDS analysis on apatite grains present in the metasedimentary rocks and identified them as fluorapatite. We have used hydroxyapatite in our experiments for simplicity. Using fluorapatite in the experiments may have some influence on the results, as it is more thermally stable than hydroxyapatite but it was not tested. We have mentioned about this in lines 270-272 the revised manuscript.

(3) Figure 6 and XRD methodology. The XRD patterns provided by the authors cannot be produced by a diffractometer using a Mo K α radiation. Instead, when I look at the patterns, a copper X-ray source is likely. The most intense peaks of quartz correspond to d-spacing s of 4.25 Å and 3.34 Å, which in turn correspond to 2θ of 20.85 and 26.70°, respectively. This is what I observed in the XRD patterns in Figures 6 and 7. Copper has a wavelength of 1.5418 Å, while molybdenum has a wavelength of 0.709 Å, which leads to different XRD

patterns. However, I thought capillary tubes were only adapted to diffractometers using a Mo K α radiation. So please clarify this point.

Response: We appreciate the reviewer's comment and acknowledge that the confusion likely arose from a lack of clarity in our original explanation.

We confirm that the XRD data were acquired using a diffractometer equipped with Mo K α radiation, which, as the reviewer correctly notes, is well suited for measurements on samples loaded in capillary tubes. However, for consistency and ease of comparison with standard reference patterns, we converted the Mo K α data to Cu K α equivalents using the PowDLL software. The values presented in the plots therefore reflect Cu K α wavelengths.

We have now clarified this procedure explicitly in the Methods section of the revised manuscript (line 478-480).

Line 492: "using Mo K α 1 radiation at room temperature from 2.5° to 40° (2 θ) with a scan rate of 2.75° (2 θ)/step". The figures show that XRD patterns were acquired from 10 to 80°2 θ . Provide the counting time per step in second.

Response: We thank the reviewer for pointing this out. The XRD data were acquired using Mo K α ₁ radiation but were converted to the corresponding Cu K α ₁ values, which are presented in Figures 6 and 7. This explains the 2 θ values shown in those figures. We have now included the total duration of each analysis, along with the scan rate, in the revised manuscript (lines 477–481).

Some XRD peaks are not indexed (example: fig. 6, purple line), why? Where do Si₂N₂O and Si₃N₄ come from? What is the source of Ni here? Because a large amount of Ni is needed to form these minerals, clarify.

Response: We thank the reviewer for pointing this out. In some of our experiments, we were unable to confidently identify certain phases. Although we compared the XRD patterns with entries in the ICSD database, no satisfactory matches were found. We have now acknowledged this limitation explicitly in the revised manuscript (captions of Fig. 6 and 7).

Regarding the reviewer's mention of "Ni," we believe this may have been a typographical error and that "N" (nitrogen) was intended. We were equally surprised to observe the formation of nitride phases such as Si₂N₂O and Si₃N₄ in a subset of our experiments. We attribute this observation to the combination of SiO₂ in the experimental setup and the high-temperature (1300 °C) conditions under a flowing N₂ atmosphere, which may have created favorable conditions for nitride formation. However, the exact mechanisms underlying the formation of these nitride phases remain unclear and merit further investigation, which is beyond the scope of the present study.

We would like to emphasize that, despite the presence of unidentified phases and the unexpected formation of nitrides in some runs, our central finding remains unaffected: carbon can act as an effective reducing agent for apatite-hosted phosphate in the presence of Fe-bearing phases, as well as for phosphate hosted in Fe-phosphate minerals.

In order to compare XRD patterns hosting barringerite from other studies, provide to the readers the following refs: Litasov et al. (2020) <https://doi.org/10.1038/s41598-020-66039-0> and Buseck (1969) <https://doi.org/10.1126/science.165.3889.169>. I recommend using the same method for schreibersite. In figure 6, change “tridymite” to “tridymite/cristobalite”. In figure 7. Check the writing of cristobalite.

Response: We have now added these references for barringerite peaks to the revised manuscript (line 244-245). We also have provided references for schreibersite XRD patterns (line 257). We have made corrections in Fig.6 and 7 as suggested.

Finally, I recommend providing a supplementary table with the most intense peaks and their corresponding d-spacing and 2θ for each mineral in order to facilitate comparisons with references. A useful ref is the book of Brindley & Brown (1980) Crystal structure of clay minerals and their X-ray identification. <https://doi.org/10.1180/mono-5>.

Response: We have added a Supplementary table & with the most intense peaks and their corresponding d-spacing and 2θ for each mineral.

(4) What about the kinetic role of the reaction? You have discussed that the chosen temperatures are in the range of rocks heated by magmatism, but the kinetic of the reaction is not addressed. Your experiments last couple of days, while the processes invoked here in natural environments could last much longer. Should this be taken into account?

Response: Since our experiments were conducted at very high temperatures, we suspect that equilibrium may have been reached. The reviewer is correct in stating that we cannot fully rule out kinetic effects. However, an important observation of our experiments is that the durations were sufficient for producing a variety reactive P species under some conditions but not others, e.g., when a mixture of apatite, silica, and carbon were heated at 1300 °C, we did not see any phosphide formation but they form when we added Fe sources in this mixture (Fig. 7). This supports the notion that the experimental conditions (specifically, the composition of reagents) rather than time were the most relevant parameter.

Line-by-line comments:

The words “**metasediments**” or “**metasedimentary**” have to be used everywhere. Please check in the manuscript.

Response: We have changed the wording whenever required as suggested to emphasize that the studied rocks are metasediments.

Lines 103-107: Does this correspond to the MdL2 unit? If so, this should be clarified.

Response: Are you referring to the term “magmatic activity”? Indeed, MdL is part of it. To clarify this point, we replaced “noteworthy event” by “conspicuous contributor to this “flare-up”” so that the former lines 94-97 now read as follows:

Syn- and post-depositional magmatic activity affected Moodies Group strata. The most conspicuous contributor to this “flare-up” is the emplacement of the Moodies lava, a

ca. 20-400 m thick complex of basaltic amygdaloidal lavas approximately midway in the Moodies stratigraphic column. It is widely overlain by dacitic air-fall tuffs dated at 3219 ± 9 Ma, 3222 ± 8 Ma, and 3228 ± 8 Ma (LA-ICP-MS U-Pb ages of zircon (17)).

Figure 1. Can all boreholes be mapped here? BASE-1A, BASE 2A, BASE 4B etc... are mentioned in the text but it's hard to visualize where they are with respect to the studied core.

Response: We added locations of the other BASE boreholes and modified legend and caption accordingly.

Lines 133-134: "magmatic-sedimentary interaction that took place during the Paleoproterozoic, prior to 3.2 Ga". Do you mean the one that occurred in the MdL2 unit?

Response: Correct. We inserted "at ca. 3.224 Ga" so that the text now reads:

Intrusion at ca. 3.224 Ga may have occurred at shallow depths and into unconsolidated sediment, as suggested by the nonlinear contacts, or occurred thousands to a few million years later, during late deformation of the BGB and subsequent beginning consolidation of the Kaapvaal craton.

Line 140: Remove "mostly" before "quartz".

Response: Done

Line 141: Fig. 2A. You have mentioned the presence of "carbonaceous laminae" in Fig. 2A, but the TOC is extremely low. The black laminae are likely due to the presence of numerous fine-grained minerals like clays.

Response: This may be possible. We have changed the term as 'carbonaceous and clay laminae'.

Line 168: remove the "s" to "yields".

Response: Done as suggested.

Line 184: "39.218", here only two digits after the decimal point for consistency.

Response: Done as suggested.

Lines 193-194: Maybe Supplementary Table **S5 and S6**?

Response: Right. Corrected.

Lines 196-198: Did you replicate the analysis and measurements? If true, I presume you will have the same P(V) concentration. The tone of the sentence shows you are very confident in this statement.

Response: Most of these experiments were conducted once, hence we have changed the language in this paragraph to indicate that minor variations may be seen in the starting material.

Line 223: Should be 0%.

Response: Right. Done as suggested.

Figure 6. In the insert of the Apa+SiO₂+CB+Fe₃O₄ NMR spectra, change the scale. Provide two digits after the decimal point.

Response: We assume that the reviewer meant Fig. 5. We have removed the scales altogether from the inset graphs mainly because they are part of the main graph and they became very small in size.

Lines 249-254: are the values of P(I) and P(III) yields significant? They are extremely low. It does not seem to me that magnesium phosphate is an important player to the formation of reduced P species. At the end of this paragraph, it is not clearly mentioned that the reduction of P(V) depends on the mineral host.

Response: We thank the reviewer for pointing this out. Our IC-ICPMS method can detect P(III) and P(I) as low as 0.1 ppb, this have allowed us to provide the % values up to three decimal points. We agree with the reviewer that the starting phosphate mineral has some effect on the reduction yield. We have made this clear in the revised manuscript (Supplementary Table S6, where the data is presented).

Line 336: "<0.004%". Could it be 0% considering the incertitude?

Response: Our IC-ICPMS method can detect P(III) and P(I) as low as 0.1 ppb, this have allowed us to provide the % values up to three decimal points. We have made this clear in the revised manuscript (Supplementary Table S6, where the data is presented).

Line 351: A schematic model could be of great interest to support the interpretation as Baidya et al. (2024) did.

Response: We have added a schematic diagram (Fig. 8 in the revised manuscript). The new figure looks like this-

Line 543: "thrice", should it be three times?

Response: Corrected.

I hope the authors find these comments helpful, and I am willing to review the manuscript again if needed.

Jérémie Aubineau

Summary: The authors present an interesting study in which they performed mineralogical and geochemical analyses of Archean metasediments combined with laboratory experiments mimicking thermal metamorphism of organic carbon- and phosphorus-bearing sediments. Supported by their experimental works, the authors reveal the likely origin of polyphosphates and reduced phosphorus species that are more soluble and reactive than phosphate ions. These findings bring new insights for deciphering the origin of life in the prebiotic world of the early Earth

I really appreciated the presented manuscript. Overall, the manuscript is well written, interesting and relatively easy to understand. The introduction and discussion sections are clear, and I did not find any major flaws in the data interpretation. However, I have major/moderate comments regarding the results and XRD methodology. They need to be addressed before publication. I hope this will not change the data interpretation.

The methodology seems appropriate to the conducted study, although I do not have any specialist knowledge on heating experiments and nuclear magnetic resonance spectroscopy. Therefore, I can only offer limited judgement in this regard. This research offers a novel viewpoint by exploring polyphosphates and reduced phosphorus species in rocks from the early Earth, and it should be of interest to readers of *Communication Earth & Environment*. As I am not a native English speaker, I would not feel comfortable taking responsibility for a full linguistic review.

Major/moderate comments:

(1) I do not understand why the authors used the geometric mean. There is no mathematical explanation. The geometric mean is usually used for calculating the mean of a consecutive percentage of when the data follow a lognormal distribution. Use 2 standard deviations around the arithmetic mean for representing the variability of the data.

ISO 5725 uses two terms “trueness” and “precision” to describe the accuracy of a measurement method. “Trueness” refers to the closeness of agreement between the arithmetic mean of a large number of test results and the true or accepted reference value. “**Precision**” refers to the closeness of agreement between test results (ISO 5725-1, 1994). Measurement precision is used to define measurement repeatability. Also, if you want to keep the geometric mean, please clearly explain, and you have to calculate the **geometric** standard deviation.

In figure 3, I recommend changing Fig 3E-H to A-D for consistency with the text.

Clarify in the caption that the data are presented in Table S4 because I thought the data were wrong as I looked at the table S3.

And finally, why are the data plotted as a function of Cr/Ti? This needs to be clarified.

(2) Apatite. In the Archean metasediments, you show the presence of apatite minerals. Are they fluorapatite or hydroxyapatite? Fluorapatite is a more common minerals in magmatic rocks. Isn't it? This is not clear in the SEM images; you could realize some EDS measurements to confirm the presence of fluorapatite or hydroxyapatite. Because I am wondering whether the experiments would have provided similar results with fluorapatite instead of hydroxyapatite.

(3) Figure 6 and XRD methodology. The XRD patterns provided by the authors cannot be produced by a diffractometer using a Mo $K\alpha$ radiation. Instead, when I look at the

patterns, a copper X-ray source is likely. The most intense peaks of quartz correspond to d-spacing s of 4.25 Å and 3.34 Å, which in turn correspond to 20.85 and 26.70 °2θ, respectively. This is what I observed in the XRD patterns in Figures 6 and 7. Copper has a wavelength of 1.5418 Å, while molybdenum has a wavelength of 0.709 Å, which leads to different XRD patterns.

However, I thought capillary tubes were only adapted to diffractometers using a Mo K α radiation. So please clarify this point.

Line 492: “using Mo K α 1 radiation at room temperature from 2.5° to 40° (2θ) with a scan rate of 2.75° (2θ)/step”. The figures show that XRD patterns were acquired from 10 to 80°2θ. Provide the counting time per step in second.

Some XRD peaks are not indexed (example: fig. 6, purple line), why? Where do Si₂N₂O and Si₃N₄ come from? What is the source of Ni here? Because a large amount of Ni is needed to form these minerals, clarify.

In order to compare XRD patterns hosting barringerite from other studies, provide to the readers the following refs: Litasov et al. (2020) <https://doi.org/10.1038/s41598-020-66039-0> and Buseck (1969) <https://doi.org/10.1126/science.165.3889.169>. I recommend using the same method for schreibersite.

In figure 6, change “tridymite” to “tridymite/cristobalite”. In figure 7. Check the writing of cristobalite.

Finally, I recommend providing a supplementary table with the most intense peaks and their corresponding d-spacing and °2θ for each mineral in order to facilitate comparisons with references.

A useful ref is the book of Brindley & Brown (1980) Crystal structure of clay minerals and their X-ray identification. <https://doi.org/10.1180/mono-5>.

(4) What about the kinetic role of the reaction? You have discussed that the chosen temperatures are in the range of rocks heated by magmatism, but the kinetic of the reaction is not addressed. Your experiments last couple of days, while the processes invoked here in natural environments could last much longer. Should this be taken into account?

Line-by-line comments:

The words “**metasediments**” or “**metasedimentary**” have to be used everywhere. Please check in the manuscript.

Lines 103-107: Does this correspond to the MdL2 unit? If so, this should be clarified. Figure 1. Can all boreholes be mapped here? BASE-1A, BASE 2A, BASE 4B etc... are mentioned in the text but it's hard to visualize where they are with respect to the studied core.

Lines 133-134: “magmatic-sedimentary interaction that took place during the Paleoproterozoic, prior to 3.2 Ga”. Do you mean the one that occurred in the MdL2 unit?

Line 140: Remove “mostly” before “quartz”.

Line 141: Fig. 2A. You have mentioned the presence of “carbonaceous laminae” in Fig. 2A, but the TOC is extremely low. The black laminae are likely due to the presence of numerous fine-grained minerals like clays.

Line 168: remove the “s” to “yields”.

Line 184: “39.218”, here only two digits after the decimal point for consistency.

Lines 193-194: Maybe Supplementary Table **S5 and S6**?

Lines 196-198: Did you replicate the analysis and measurements? If true, I presume you will have the same P(V) concentration. The tone of the sentence shows you are very confident in this statement.

Line 223: Should be 0%.

Figure 6. In the insert of the Apa+SiO₂+CB+Fe₃O₄ NMR spectra, change the scale. Provide two digits after the decimal point.

Lines 249-254: are the values of P(I) and P(III) yields significant? They are extremely low. It does not seem to me that magnesium phosphate is an important player to the formation of reduced P species. At the end of this paragraph, it is not clearly mentioned that the reduction of P(V) depends on the mineral host.

Line 336: "<0.004%". Could it be 0% considering the incertitude?

Line 351: A schematic model could be of great interest to support the interpretation as Baidya et al. (2024) did.

Line 543: "thrice", should it be three times?

I hope the authors find these comments helpful, and I am willing to review the manuscript again if needed.

Jérémie Aubineau